# Biomimetic KcsA channels with ultra-selective K$^+$ transport for monovalent ion sieving

Weiwen Xin [1,2,4], Jingru Fu[3,4], Yongchao Qian[1], Lin Fu[1,2], Xiang-Yu Kong [1], Teng Ben [3✉], Lei Jiang[1,2] & Liping Wen [1,2✉]

Ultra-selective and fast transport of K$^+$ are of significance for water desalination, energy conversion, and separation processes, but current bottleneck of achieving high-efficiency and exquisite transport is attributed to the competition from ions of similar dimensions and same valence through nanochannel communities. Here, inspired by biological KcsA channels, we report biomimetic charged porous subnanometer cages that enable ultra-selective K$^+$ transport. For nanometer to subnanometer scales, conically structured double-helix columns exhibit typical asymmetric transport behaviors and conduct rapid K$^+$ with a transport rate of 94.4 mmol m$^{-2}$ h$^{-1}$, resulting in the K$^+$/Li$^+$ and K$^+$/Na$^+$ selectivity ratios of 363 and 31, respectively. Experiments and simulations indicate that these results stem from the synergistic effects of cation-$\pi$ and electrostatic interactions, which impose a higher energy barrier for Li$^+$ and Na$^+$ and lead to selective K$^+$ transport. Our findings provide an effective methodology for creating in vitro biomimetic devices with high-performance K$^+$ ion sieving.

[1] Key Laboratory of Bio-inspired Materials and Interfacial Science, Technical Institute of Physics and Chemistry, Chinese Academy of Sciences, 100190 Beijing, PR China. [2] School of Future Technology, University of Chinese Academy of Sciences, 100049 Beijing, PR China. [3] Department of Chemistry, Jilin University, Changchun 130012, PR China. [4]These authors contributed equally: Weiwen Xin, Jingru Fu. ✉email: tben@jlu.edu.cn; wen@mail.ipc.ac.cn

The advent of nanostructured materials and nanofabrication technologies has led to the development of artificial nanochannel membranes with high efficiency and exquisite ion and molecule selectivity[1–3]. However, while both bottom-up and top-down approaches can yield nanochannels with dimensions comparable to those of their natural counterparts, duplicating their affinity and transport behaviors remains challenging[4–9]. For example, the natural potassium ion channel KcsA achieves a $K^+$ ion permeability of $10^8$ ions $s^{-1}$ per channel —that is, equivalent to 6 mol $m^{-2}$ $h^{-1}$, and a monovalent ion selectivity of $K^+/Na^+$ as high as $10^4$ (ref. [10]). The ultrahigh permeability and selectivity are attributed to the asymmetrical channel morphology and unique chemistries across the continuous angstrom-scale and nanoscale filters[11–14]. Inspired by multifunctional natural counterparts, bottom-up subnanometer-size porous membranes that deliver a $Na^+$ ion transport rate of $1.3 \times 10^{-4}$ mol $m^{-2}$ $h^{-1}$, albeit with negligible $K^+/Na^+$ selectivity, have been reported[15]. Furthermore, solid-state nanopore decorated with 18-crown-6 ether is capable of high $K^+/Na^+$ selectivity which opens promising avenues toward the exquisite ion selectivity and is key for achieving artificially physiological events[9]. Additionally, artificial solid-state metal-organic frameworks that mimic the functionality of biological ion channels have been fabricated; however, the $Li^+/K^+$ and $Li^+/Na^+$ selectivity ratios were limited to 2.2 and 1.4, respectively, with $\sim 10^6$ ions $s^{-1}$ for composite pores[16]. On the other hand, recent advances in synthetic analogs fabricated using top-down methods are expected to replicate the exquisite selectivity observed in biology[17]. Recently, angstrom-sized (i.e., $\sim 9$ Å) ion channels in graphene oxide (GO) laminates have been employed to facilitate permeation up to $10^{-3}$ mol $m^{-2}$ $h^{-1}$, although the ion selectivity was less than 1 (ref. [4]). In principle, the swelling of the GO membrane reduces the ion selectivity; therefore, the capillaries, which are arranged at fixed intervals, greatly increase the $K^+/Mg^{2+}$ selectivity to over 600 (ref. [5]). Nevertheless, the $K^+/Na^+$ and $K^+/Li^+$ selectivity remains lower than 2, owing to the similar interactions between the monovalent ions and capillary walls[18,19].

Some studies have also confirmed that porous materials such as metal organic frameworks (MOFs) and covalent organic frameworks (COFs) with angstrom-sized windows are attractive candidates for ion sieving and achieve a mono-/di-valent ion selectivity that can reach as high as four orders magnitudes owing to the exit-entry effects, size (steric) exclusion, and surface charge (Coulomb force)[20–24]. Unfortunately, to date, there are few studies on the design and fabrication of two-dimensional materials with biomimetic channels or pores for mono-/mono-valent ion separation, because such ions, in particular for $Li^+$, $Na^+$, and $K^+$, exhibit similarities in some respects, such as hydration energies below 123 kcal $mol^{-1}$ and ion radii ranging from 0.6 to 1.3 Å (refs. [5,25,26]). Their similar affinities make it difficult to govern the overall transport order of different metal ions and the mechanism of the selective transport in nanochannels lacks experimental evidence[27]. In addition to MOF or COF channels, porous organic salt materials with abundant functional groups have polar pores containing water molecules consisting of nanometer-sized windows and subnanometer-sized cavities, ideal for biomimetic ion filters[28,29]. Moreover, our recent study demonstrated a permanently porous crystalline material with tailored double-helix columns of electrostatic charges that governed the proton or molecule transport behaviors[28]. In contrast with most reported porous materials, the continuous screwing cavities, that is pores, endow unique transport dynamics along the nanometer to subnanometer scales with step-wise mechanism which was actuated by specific host-guest interactions, leading to selective monovalent transport and sieving. Generally, solid-state nanochannels, far more robust than lipid bilayer, are in favor of tunable

geometry and surface functionalization, allowing more direct experiments on ion transport mechanism, wider electrochemical explorations, and easier integrations into in vitro biomimetic devices, and thus an effective strategy to establish ultra-selective and fast ion channels is highly demanded.

Herein, we show that a porous crystal, namely crystalline porous organosulfonate-amidinium salts (CPOS), can be filled into a single asymmetrical nanochannel via in situ growth to form selective ion channels that enable selective $K^+$ transport with functionality analogous to KcsA potassium channels. The biomimetic $K^+$ channels demonstrate ultrahigh $K^+$ transport rates of up to $9.44 \times 10^{-2}$ mol $m^{-2}$ $h^{-1}$, which is two orders of magnitude higher than the $Na^+$ and $Li^+$ values. Therefore, the $K^+/Li^+$ selectivity was calculated to exceed 363, making great progress in monovalent ion sieving compared with most artificial nanochannel membranes reported thus far[17,18,30–33]. Simulations and experiments demonstrate that ion-channel effects, including both cation-$\pi$ and electrostatic interactions, raise the energy barrier for $Na^+$ and $Li^+$, thereby facilitating high $K^+$ ion permeation and mobility. This synergistic effect might endow ions with brand-new transport dynamics. Moreover, we demonstrate that external biases can be applied to control the ion conductance and selectivity of biomimetic $K^+$ channels, thereby establishing a promising platform that offers controllable transport for engineering of desirable selectivity and transport characteristics at the sub-1-nm level.

## Results and discussion

**Design and in situ synthesis of biomimetic ion channels.** The concept of the $K^+$ channel KcsA, which enables rapid and selective $K^+$ ion transport in a single file across the plasma membrane[1,34], relies on the formation of asymmetric cavities with functional sites[35], as shown in the left panel of Fig. 1a. Because of the steric hindrance and interactions of dehydrated $K^+$ ions within the selectivity filter, high ion conduction is achieved via an exquisite channel mediator. Accordingly, the construction of biomimetic filters with KcsA-like $K^+$ ion channels is expected to achieve $K^+$ ion separation. To achieve this, considerable effort has been devoted to fabricating materials with high nanoporosity and confined sub-1 nm spacings, as well as suited functional surfaces for ion transport control. As demonstrated in our previous report[29], self-assembled CPOS pores, like natural KcsA, feature high ion conductivity and easily confinable polar molecules, for example, water. In the middle panel of Fig. 1a, the tetragonal $I4_1/a$ space group (Supplementary Fig. 1), a quarter of which comprises 4,4′,4″,4‴-methane tetrabenzenesulfonate (TBS), with a further half consisting of 1,4-diamidiniumbenzene (DAB) shows the unique features of the $K^+$ channel structure (see the left panel of Fig. 1a). Furthermore, the sequential channels, which have a cross section of $5.3 \times 6.8$ Å$^2$ (the right panel of Fig. 1a), have a multifunctional-walled surface for cation transport along the exposed $SO_3^-$ ribbons (green arrows).

In order to mimic the cell membrane, we created a conical transmembrane nanochannel across a post-track-etching polyimide (PI) membrane (Fig. 2b), and full details of the track-UV technique and chemical etching procedure are presented in the Methods section. The initial inner surface was rich carboxyl ($-COO^-$) groups (left inset)[36], which were subsequently activated by ethanediamine (Supplementary Methods), thus, yielding a functionalized nanochannel (step 1, middle inset of Fig. 2b)[37]. An in situ synthesis strategy was developed to assemble the CPOS material into the aforementioned single conical nanochannel, resulting in nanometer-to-subnanometer columns for ionic transport (step 2, right inset of Fig. 2b, Supplementary Fig. 2).

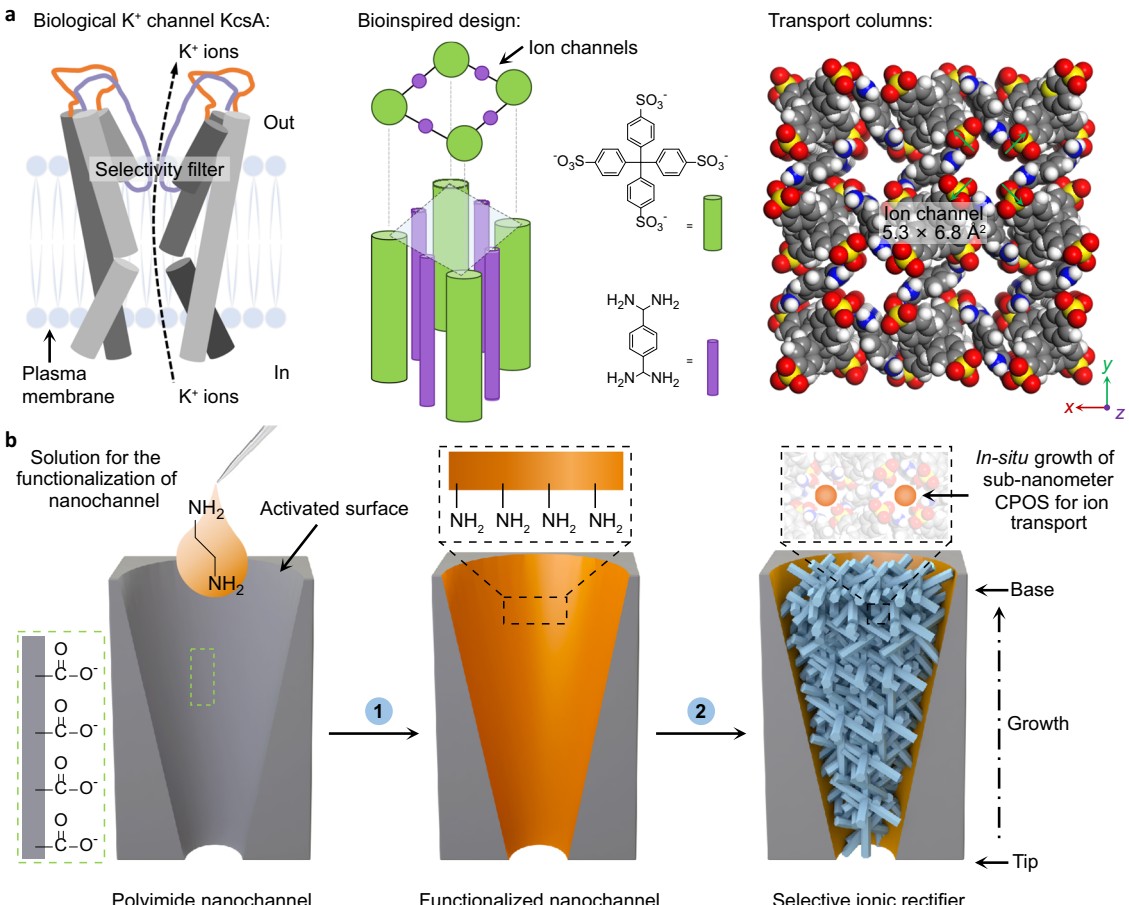

**Fig. 1 Design and synthesis of the biomimetic K⁺ channels. a** KcsA K⁺ channels embedded in the plasma membrane (left panel) inspired our biomimetic K⁺ channels composed of CPOS units (middle panel). The empty channels with a cross section of 5.3 × 6.8 Å² (right panel) provide multifunctional channels for ion transport. The green arrows indicate the exposed $-SO_3^-$ ribbons (C, gray; N, blue; O, red; S, yellow; H, white). **b** Schematic showing the preparation and incorporation of the CPOS into conical transmembrane nanochannels. A single conical nanochannel ($-COO^-$ wall) was activated using ethanediamine molecules, which provide a functionalized inner surface on the post-track-etching polyimide (PI) membrane (step 1). Subsequently, in situ growth occurred in which the nanoparticle seeds, namely CPOS (see Supplementary Methods for details), were assembled from the tip to the base of the nanochannel (step 2).

The scanning electron microscopy (SEM) image in Fig. 2a depicts the nanocone, which has an average inner base diameter of 750 nm and a tip diameter of 75 nm (Supplementary Fig. 3). Specifically, the TBS seeds were first riveted onto the functionalized surface, inducing self-assembly in the presence of DAB[28]. As shown in Fig. 2b, the synthesized CPOS filled the nanocone without distinct defects, which is attributed to the in situ heteronucleation during the crystallization of extended frameworks[38]. In addition, thermogravimetry confirmed that the CPOS remained thermally stable until 600 K (Supplementary Fig. 4). The powder X-ray diffraction (PXRD) patterns in Fig. 2c confirm that the in situ strategy gives rise to highly crystalline and stable CPOS (black spot) in the confined nanochannel, thus in good agreement with the previously reported data and calculated result[28], which is different from the subtle information of the PI surface (orange spot). Meanwhile, the $CO_2$ adsorption-desorption profile of the CPOS (measured at 273 K) resembles a type-I isotherm followed by a sharp uptake at low pressures ($P < 0.10$), which is characteristic of microporous materials (Fig. 2d). The total pore volume at $P = 0.90$ was calculated as $V_p = 0.081$ cm³ g⁻¹. The pore size distribution of the CPOS was calculated using non-local density functional theory, indicating pores sizes of alternative 7.9 Å and 10.5 Å (Fig. 2e), thereby supporting the proposed nanometer-to-subnanometer model. The large proportion of $-SO_3^-$ groups on the inner surface of the

confined nanochannels imparts a negative charge that promotes fast cation transport[39], which is confirmed by a zeta potential of −29.3 mV, as shown in Fig. 2f. Beyond that, the charged and subnanometric structural features offset the influence of the anions, for example, $Cl^-$.

**Selective cation transport via CPOS pores.** To gain deeper insights into the ion transport properties across the fabricated conically structured nanochannel, the applied voltage was increased in a stepwise manner, allowing the current-voltage vs Ag/AgCl relationships to be monitored[40] (see Methods and Supplementary Fig. 5). In Fig. 3a, the rectified I–V curve for the unmodified single nanochannel reflects its conical shape and negative surface charge, in accordance with the characteristics of the pristine nanochannel in Fig. 1b. Importantly, the pristine nanochannel lacks the ion-selective transport (Supplementary Fig. 6) owing to the similar affinities of different ions in the single nanochannel. In the case of the nanochannel modified with CPOS, the current decreased from −83.6 to −41.3 nA at −1.0 V and from 26.2 to 0.1 nA at 1.0 V; however, the calculated rectification ratio of the functional nanochannel (see inset of Fig. 3a and Supplementary Methods) was three times higher than that of the unmodified nanochannel (9.7 rather than 3.2). This is mainly

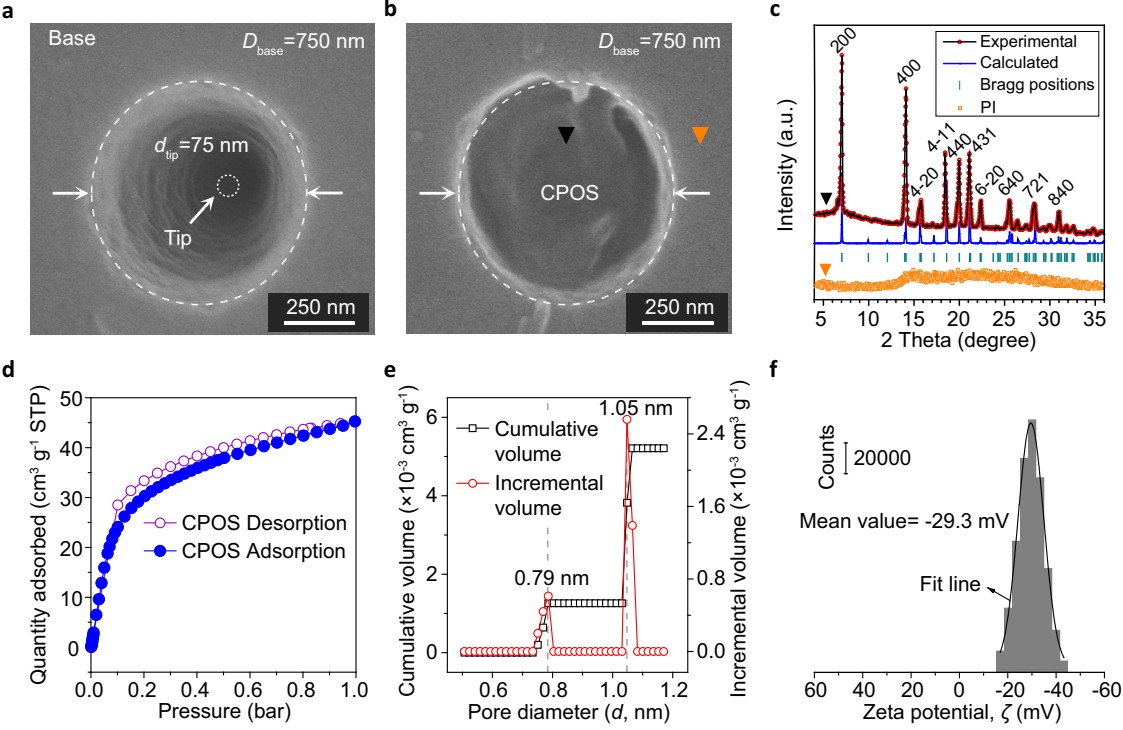

**Fig. 2 Characteristics of the functionalized nanochannel. a** SEM image of the base and tip profiles of the asymmetrical nanochannel. The nanocone has an average inner base and tip diameter of 750 nm and 75 nm, respectively (Supplementary Methods). **b** SEM image showing the in situ synthesis of the CPOS material, which is full of the nanocone due to the initial growth from the tip to the base. **c** PXRD pattern of CPOS, demonstrating their successful embedding within the confined nanochannel. This further confirmed two different locations on the surface (black and orange spots), as indicated in the XRD profiles. **d** The $CO_2$ adsorption-desorption isotherms of the CPOS exhibit type-I isotherm with a sharp uptake at low pressures ($P < 0.10$). **e** Pore size distributions of the CPOS, showing the size variation for the nanometer-to-subnanometer channels. **f** Surface zeta electric potential of the doubly helical columns, which have net negative charge.

attributed to the increase in negative surface charge and hydrophilicity on the inner surface, which promotes enhanced cation selectivity[41]. As such, the increased asymmetric $I–V$ relationship, which depends on the angstrom-scale functionalization, indicates the feasibility of selective ion transport through CPOS pores. In addition, ionic rectification behaviors and voltage-activated ion flux are typically related to the shape of the underlying potential within the nanochannel[42], which arises from ion hydration, cation-π, and electrostatic interactions[26], regardless of whether the channels are biological or synthetic.

Consequently, conically structured CPOS pores exhibit remarkable ion selectivity, with the rectification ratios for five types of cations ($Li^+$, $Na^+$, $K^+$, $Mg^{2+}$, and $Ca^{2+}$) presented as a bar plot in Fig. 3b. The CPOS pores exhibited repeatable behaviors, further indicating preferential transport of $K^+$ over $Li^+$, $Na^+$, $Mg^{2+}$, and $Ca^{2+}$. In addition to observing similar $K^+$ selectivity trends in 0.1 and 0.01 M feed solutions, we also found that the rectification in the 0.01 M solution is almost two times higher than that in the 0.1 M solution, owing to the surface-charge-governed ion transport inside the channel, which fully deviates from the bulk behaviors[43] (dashed line) when the concentration is < 0.1 M (Fig. 3c). The affinity of the concentration ($c_b$) and conductance ($G_0$) in a single CPOS pore can be qualitatively described as $G_0 = 10^3(\mu_+ + \mu_-)c_b N_A ewh/l + 2\mu_+ \sigma w/l$ (refs. [43,44]), where $\mu$, $N_A$, and $e$ represent the ion mobility, Avogadro constant, and elementary charge, respectively; $w$ and $l$ are the width and length of a single CPOS pore, respectively; $h$ is the height of a CPOS cavity, and $\sigma$ is the surface charge density. In high-concentration region, the ion conductance is linearly proportional to $c_b$ with the $R^2 = 0.995$ that is in good agreement

with the bulk term of $10^3(\mu_+ + \mu_-)c_b N_A ewd/l$. Moreover, the conductance in low-concentration region, also defined as $2\mu_+ \sigma w/l$, comes from the contribution of the ions accumulated in strong Debye layer overlaps. In that case, a conductance plateau insensitive to the solution concentration occurs in which the surface charge dominates ion transport. Benefiting from this ion transport control, the conductance in the 0.01 M solution was 26.9 nS, only marginally higher than 27.5 nS in the 0.1 M solution. Similarly, the nonlinear ion conductance across the CPOS confirmed the existence of charge around the surface of the pores. Highly charged ions with a large radius, such as $Mg^{2+}$ and $Ca^{2+}$, generate a large transport resistance (Supplementary Table 1), resulting in low ionic flux. Importantly, the contrast between monovalent and bivalent metal ions indicates the significant role of the CPOS pores as monovalent ion transport filters, especially for $K^+$, $Na^+$, and $Li^+$ (Supplementary Fig. 7). Such artificial channels that offer selective $K^+$ transport provide a platform for realizing ion sieving among monovalent ions.

The cycling performance of $K^+$ channels manifests in the $I−V$ response of the asymmetrically structured CPOS pores. As shown at the top of Fig. 3d, the $I−V$ curves exhibit good stability and responsive switching upon converting the external voltage from −1.0 to 1.0 V (Fig. 3d, bottom). Notably, in the case of applied negative bias, the $SO_3^-$ ribbons and abundant phenyl groups inside the asymmetric CPOS pores contribute to enhancing the interactions from the electrostatic and cation-π effects between ions and the inner surface of the nanochannels[25,45], resulting in increased transmembrane ionic flux. Meanwhile, the ion rectification effect of three cycles (namely, cycles 2, 7, and 11) in Fig. 3d was observed to be stable for asymmetrical ionic

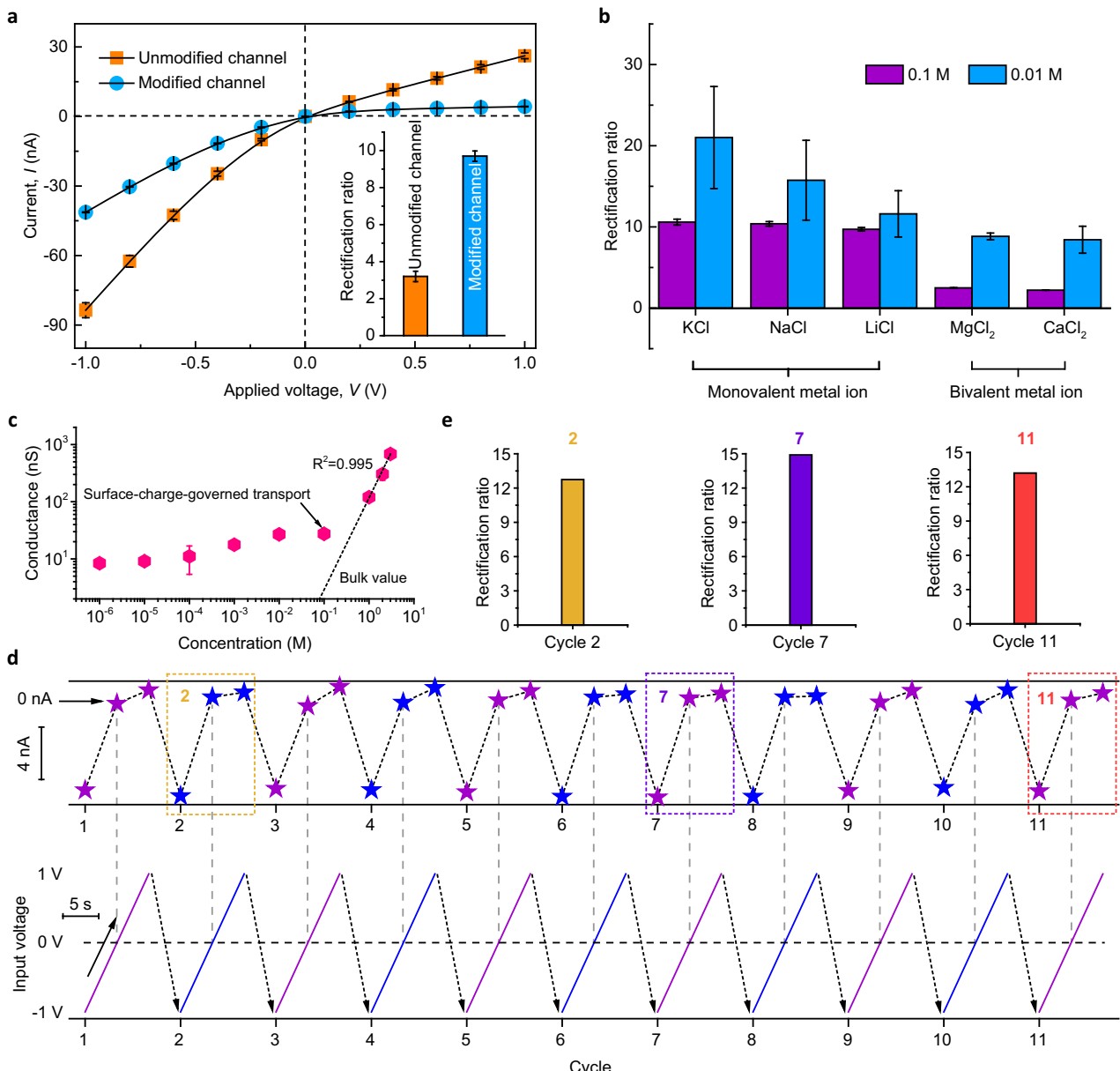

**Fig. 3 Asymmetric ion transport behaviors of the biometric K$^+$ channels. a** $I$–$V$ curves of the conical nanochannel before (orange) and after (blue) CPOS growth, respectively, measured in 0.1 M solutions (pH 6.8). Inset, the corresponding ion rectification, as calculated by $|I_-|/|I_+|$. Error bars denote the standard deviation. **b** Bar plot summarizing the rectification ratios of monovalent (K$^+$, Na$^+$, and Li$^+$) and bivalent metal ions (Mg$^{2+}$ and Ca$^{2+}$) through the modified nanochannel in 0.1 and 0.01 M solutions, respectively. Error bars exhibit the standard deviation. **c** Transmembrane ion conductance as a function of the solution concentration. The experimental conductance decreases nonlinearly with the decrease in KCl concentration, indicating the existence of surface-charge-governed ion transport. The black, dashed fit line ($R^2 = 0.995$) shows the remarkable deviation from the bulk value when the concentration is <0.1 M. Error bars represent standard deviation ($n = 11$). **d** Cycling performance of the biomimetic K$^+$ channels. The asymmetric ionic transport properties (top) of the conical nanochannel were investigated under a symmetric voltage of ±1 V (bottom). **e** Three stochastic cycles (namely cycles 2, 7, and 11), with the calculated rectification ratios showing the stable asymmetric ionic transport.

transport (Fig. 3e) owing to the compact growth of the CPOS material in a single nanocone.

**Mechanism of selective K$^+$ ion transport.** In aqueous environments, the binding behaviors of metal ions to the inner surfaces of CPOS pores are crucial for determining the mechanism of selective ion transport. The unique screw-wise transport of ions traveling along the biomimetic columns shown in Fig. 1a with step-wise reorientation dynamics is driven by the aforementioned electrostatic ($-SO_3^-$ ribbons) and cation-π (phenyl groups) interactions. The combination of ions with the CPOS was

confirmed by high-resolution X-ray photoelectron spectroscopy (XPS) measurements (Fig. 4a–c). The samples displayed characteristic peaks with binding energies at 55.8 eV (Li 1$s$), 1072.1 eV (Na 1$s$), and 293.6 eV (K 2$p$), thereby confirming the binding behaviors of ions to the inner surfaces (Supplementary Methods). No corresponding peaks were observed in the pristine CPOS samples (Supplementary Fig. 8). This result indicates that the metal ion agglomerates on the surface and the pores inside the CPOS material contribute to the ion flux increase, and thus facilitate ion diffusion. Further investigations into the structural changes of the CPOS were evaluated via $^1$H nuclear magnetic

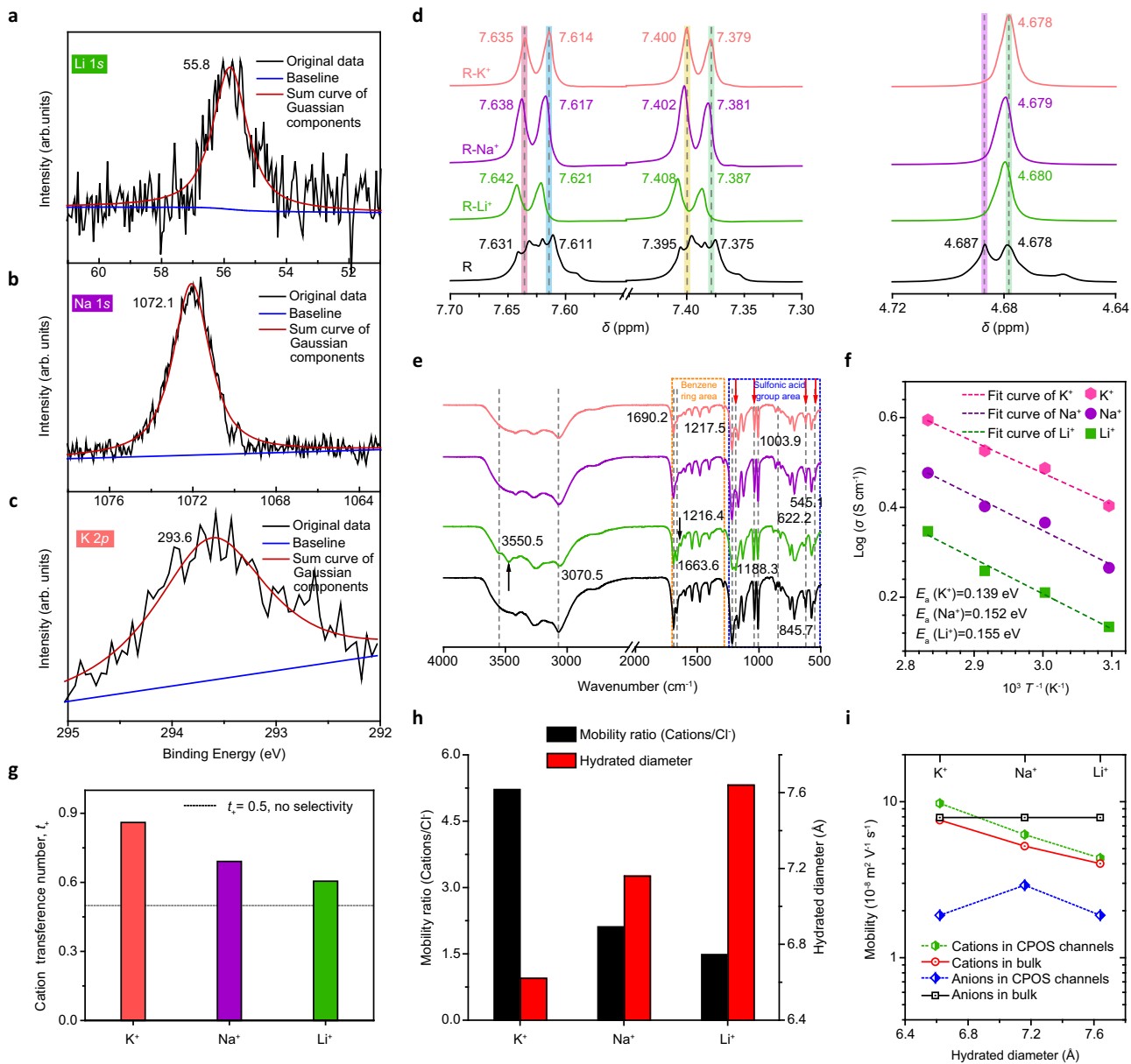

**Fig. 4 Mechanism of selective K⁺ transport.** High-resolution Li 1s (**a**), Na 1s (**b**), and K 2p (**c**) XPS spectra for the CPOS material. Samples were treated using corresponding halide solutions and then washed thoroughly. The obvious peaks with binding energies of 55.8 eV (Li 1s), 1072.1 eV (Na 1s), and 293.6 eV (K 2p) confirmed the interactions between the metal ions and the CPOS material, which were not observed in pristine CPOS samples (Supplementary Fig. 8). **d** Partial ¹H NMR spectra (400 MHz, 298 K) of R in the presence of various metal ions showing the proton resonance shifts for Li⁺, Na⁺, and K⁺, with TMS as an internal standard set at 0 ppm. The addition of metal ions led to upfield shifts with respect to the reference peak (black), with the extent of the shift indicating the corresponding interactions. **e** FTIR spectra of the CPOS (black) with Li⁺ (green), Na⁺ (purple), and K⁺ (pink). The characteristic peaks at 3550.5, 1690.2, 1188.3, and 545.1 cm⁻¹ represent distortions and transformations due to the interactions with metal ions. **f** Temperature dependence of the ion conductivity for of Li⁺ (green square), Na⁺ (purple circle), and K⁺ (pink hexagon). Solid symbols represent experimental data; dashed lines represent curve-fitting results. **g** Cation transference number through the negatively charged CPOS pores. **h** Mobility ratio of metal ions (cations/Cl⁻) as a function of cation radii. **i** Mobility of metal ions through the CPOS pores. The net negative charge of the channels facilitates cation transport.

resonance (NMR) measurements (Fig. 4d) when ions approached and entered the CPOS pores. The R (that is, TBS) in D₂O shows quadruple resonance peaks at ~7.62 and ~7.40 ppm, which are assigned to the signals from ArH and D₂O, and double resonance peaks at ~4.68 ppm, which are attributable to −CH₂. With the addition of metal ions (at a molar ratio of 4:1 for ions/R), consistent upfield shifts in the axle were observed with respect to the R reference, possibly caused by the shielding effect of the coordinating donor atom electron density of R around the metal ions.

Interestingly, the comparison of these spectra reveals that, upon exposure to K⁺ only, no distinct shift (0.04 ppm) occurred, whereas for Li⁺ and Na⁺, the related resonances produced larger upfield shifts (0.1 ppm). In agreement with the $I-V$ characteristics, the CPOS pore is suitable for rapid and selective K⁺ ion transport. Fourier-transform infrared spectroscopy (FTIR) of the samples revealed strong bands at 3550.5, 1690.2, 1188.3, and 545.1 cm⁻¹, which can be assigned to phenyl (yellow dashed frame) and sulfonic acid (blue dashed frame) vibrations[28], as

shown in Fig. 4e (for details see Supplementary Fig. 9). The positions and relative intensities of these peaks for R−Li$^+$, −Na$^+$, and −K$^+$ are different from R in the FTIR spectrum, with the changes for R−K$^+$ showing the least difference, further confirming that K$^+$ ions require the least energy.

Based on temperature-dependent conductivity profiles (Fig. 4f, Methods), the ion transport activation energy ($E_a$) was determined to be 0.155, 0.152, and 0.139 eV for Li$^+$, Na$^+$, and K$^+$ ions, respectively (Supplementary Methods), which is superior to results reported previously[21,32,33]. This low transport barrier implies an ion-hopping mechanism for diffusion in the crystal pores. Although the size of the first hydration shell ($R_{min}$) of K$^+$ is larger than those of Li$^+$ and Na$^+$ (Supplementary Fig. 10 and Supplementary Table 1), the high ion flux and $I$−$V$ response are consistent with the lower hydration energy of K$^+$ ions[26] (Supplementary Table 1). Qualitatively, the enhancement of K$^+$ selectivity via negatively charged CPOS pores can be explained using the cation transference number, $t_+$. In Fig. 4g, the $t_+$ of K$^+$ is 0.860, far higher than the 0.605 and 0.691 obtained for Li$^+$ and Na$^+$, respectively, indicating that the channel indeed enables selective K$^+$ transport. Ion diffusion dynamics were demonstrated using the drift-diffusion technique[20,31] (Methods). The mobility ratio is defined as $\mu_+/\mu_-$ (ref. [44]), that is, cation/Cl$^-$ (for more details see Methods). In the case of Cl$^-$ paired with Li$^+$, Na$^+$, and K$^+$, the K$^+$/Cl$^-$ selectivity was approximately three times that of Li$^+$/Cl$^-$ and Na$^+$/Cl$^-$, as plotted in Fig. 4h. Furthermore, the ion conductivity is expressed as $\sigma = 10^3 F(c_+\mu_+ + c_-\mu_-)$ (ref. [20]), where $c_+$ and $c_-$ are the cation and anion concentrations, respectively, and $F$ is the Faraday constant. Accordingly, the calculated metal ion conductivity in the CPOS material exceeds that in bulk (Fig. 4i), likely owing to the unique ion-hopping mechanism that accelerates ions through the channels. The $\mu_+$ value is obtained using the mobility ratio and ion conductivity. A peak K$^+$ mobility of $9.76 \times 10^{-8}$ m$^2$ V$^{-1}$ s$^{-1}$ was obtained, exceeding previously reported values[8,20,21], while the Li$^+$ and Na$^+$ mobilities were calculated to be $4.35 \times 10^{-8}$ m$^2$ V$^{-1}$ s$^{-1}$ and $6.15 \times 10^{-8}$ m$^2$ V$^{-1}$ s$^{-1}$, respectively. Overall, the stacking screwing cavities form continuous pathways for ion transport with hopping dynamics along the nanometer-to-subnanometer spacing by step-wise mechanism which was actuated by specific host-guest interactions, and thus leads to fast and selective K$^+$ transport.

**Ultrahigh K$^+$/Li$^+$ and K$^+$/Na$^+$ sieving.** Across biological ion channels in the cell membrane, ions are able to undergo gradual dehydration by translocating into the asymmetric filter (Fig. 1a, left panel), whereas in biomimetic analogs, ions are first adsorbed onto the pore surface and lose most of the outer hydration shells, which is necessary to overcome the hydration energy barrier required to enter their nanochannels[46]. As a result, the distinct transport characteristics of different ions arise, in part, from the kinetics of hydrogen bond formation and breakage[3]. Molecular dynamics (MD) simulations yield in-depth information on the process, with Fig. 5a depicting the radial distribution functions (RDFs), $g(r)$, of water molecules around the cations in the center of the screwing cavity[47,48] (Supplementary Fig. 11). All the oxygen distribution profiles, $g_{ion-O}$, of hydration ions (Fig. 5a, left) display the accumulation of oxygen atoms inside the hydration shells as opposed to hydrogen atoms (Fig. 5a, right). Moreover, the $g_{ion-O}$ RDFs illustrate the position of the oxygen density around K$^+$, which is much lower than that around Li$^+$ and Na$^+$, supporting the weaker hydration energy for K$^+$. Compared with the results in bulk water, the distance of oxygen atoms from ion becomes longer, suggesting the interaction between hydrated water molecules and the walls of the CPOS pores. In this case, hydrogen atoms are more closed to ions due to the deformation of these hydrated water molecules (Supplementary Fig. 12). In particular, careful inspection of the atomic density curves in Fig. 5b reveals the

emergence of two peak positions along the z-axis, with the peak intensity increases gradually when changing cations from K$^+$ to Li$^+$ (Fig. 5b). In addition, the hydrogen density profiles indicate a considerable hydrogen density close to the CPOS surface and represent direct ion-channel interactions. In Fig. 5b, Li$^+$ ion shows the hydration shell, which is easy to be confined in the CPOS pores and is not contributed to fast transport. The hydration shell of K$^+$ ion is softer and easier to dehydrate and rapid transport in the CPOS pores. The resultant changes in the low hydrogen density near the CPOS material suggest that the formation of hydration shells around ions plays a crucial role in the hopping process for ion transport.

Figure 5c shows the transport rate as a function of hydration free energy. The transport rate of K$^+$ reaches $9.44 \times 10^{-2}$ mol m$^{-2}$ h$^{-1}$ and was one to two orders of magnitude higher than the corresponding Na$^+$ and Li$^+$ values. It is evident from these results that K$^+$ has a lower hydration energy (Fig. 5c, pink dashed line), smaller hydration diameter, and weaker interaction with the channel surface (Fig. 4) than Na$^+$ and Li$^+$, which both form larger transport barriers in the CPOS pores. Importantly, these experimental findings are also consistent with our MD simulations. For comparison, the ion concentration in the permeation compartment was measured using inductively coupled plasma mass spectrometry, leading to a K$^+$ concentration estimated of 185.63 µg ml$^{-1}$, far higher than the 5.88 and 0.51 µg ml$^{-1}$ estimated for Na$^+$ and Li$^+$, respectively (Fig. 5d, left and Supplementary Fig. 13). Thus, the monovalent ion selectivity was calculated to be 363.8 and 31.6 for K$^+$/Li$^+$ and K$^+$/Na$^+$, respectively (Fig. 5d, right), which supports the ultrahigh K$^+$ selectivity and K$^+$ ion flux in biomimetic CPOS pores. Furthermore, owing to the differences in transport barrier and ionic radii, the channels also exhibit Na$^+$/Li$^+$ selectivity, which is in accordance with the simulations. Although the selective K$^+$ channels remain no match for their natural counterparts, the CPOS pores achieve an excellent balance between transport rate and selectivity utilizing both the cation-π interactions between ions and aromatic compounds as well as the electrostatic interactions between ions and sulfonate groups.

The ultrahigh K$^+$ channels also exhibit stable cyclic ion selectivity, with a K$^+$/Li$^+$ selectivity in excess of 300 maintained, as depicted in Fig. 5e. As an emerging two-dimensional material, CPOS not only provides typical confined spacings for ion transport, but more importantly, multifunctional surfaces that favor the ion-hopping mechanisms and synergistic effects that facilitate exclusive ion selectivity in screwing cavities. To the best of our knowledge, the selectivity between monovalent metal ions in CPOS pores, especially for K$^+$/Li$^+$, is between 10 and 100 times higher than that most reported using artificial nanopore/channels or membranes in previous studies[5,17–20,30,31,49,50] (Fig. 5f and Supplementary Table 2). Besides, the selectivities of three kinds of monovalent ion sieving demonstrate an outstanding balance which suggests wider applications than other materials. As shown in Supplementary Fig. 14, the CPOS pores also achieved the ultrahigh selectivity of K$^+$/Rb$^+$ and K$^+$/Cs$^+$ which is up to ~110 and ~1300, respectively, owing to the larger ion radius and radius of the first hydration shell of Rb$^+$ and Cs$^+$. This suggests that the CPOS pores could show a wide application in separation of the radioactive elements such as cesium isotopes. Our crystalline pores share subnanometric structural features analogous to biological ion channels, most notably their high selectivity and ultrafast transport through the high-density pores of the CPOS; as such, they demonstrate great promise for developing ion sieving devices. In fact, we expect that, by regulating the aromatic and sulfonate group content, it is possible to tune the selectivity and realize the demand for real-world K$^+$ ions extraction devices that compete with their natural counterparts.

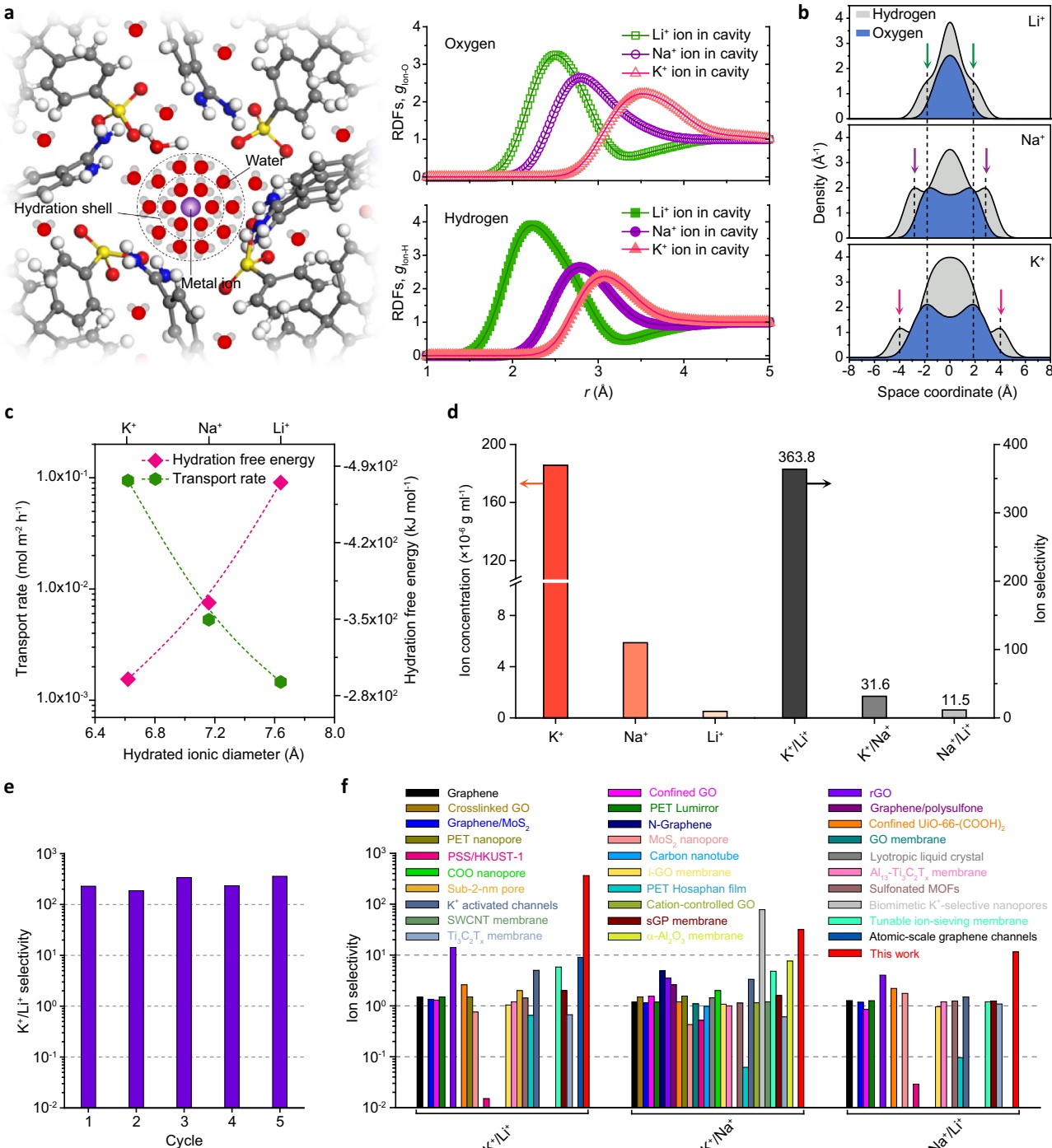

**Fig. 5 Biomimetic K$^+$ ion channels with ultrahigh K$^+$/Li$^+$ and K$^+$/Na$^+$ selectivity. a** Theoretical RDFs of steady-state ion–oxygen and ion–hydrogen distances in the CPOS pores (Supplementary Fig. 11). **b** Hydrogen and oxygen atomic density profiles along the z-axis for Li$^+$ (top panel), Na$^+$ (middle panel), and K$^+$ (bottom panel) around the screwing cavity. **c** Experimental ion transport rate as a function of the hydration ionic diameter and hydration free energy, respectively. The transport rate through the CPOS pores was measured using ternary ion mixtures as feed solutions and an applied voltage of –1 V (Supplementary Fig. 13). The green and pink dashed lines represent the fitting results for the transport rate and hydration free energy data, respectively. **d** Comparison of the ion concentration in the permeation compartment (left). The feed solutions include a ternary ion mixture of LiCl, NaCl, and KCl, while the permeation compartment was filled with deionized water (see Methods and Supplementary Methods for details). As a result, ion selectivity in the CPOS channels was obtained by calculating the ion concentrations ratios in the permeation compartment (right). **e** The K$^+$/Li$^+$ selectivity was performed for five cycles by conducting independent measurements. Results showed that the selectivity remained ultrahigh and that stable monovalent ionic separation was maintained. **f** Comparison of the K$^+$/Na$^+$, K$^+$/Li$^+$, and Na$^+$/Li$^+$ ion selectivity performance reported using nanopore/channels or membranes (Supplementary Table 2).

In summary, we successfully demonstrated the in situ growth of biomimetic sub-1-nm channels with double helices of electrostatic charges and a controllable structure that allows ultrahigh $K^+$ transport and $K^+/Li^+$ and $K^+/Na^+$ selectivity ratios exceeding ~350 and 30, respectively. The ability to achieve the effective and selective transport of metal ions in the CPOS pores via a controllable external bias, particularly at the sub-nanometer level, facilitates directional and fast diffusion of ions through the nanochannel. Taken together, our experiments and MD simulations indicate that the high $K^+$ transport and selectivity arise from the low transport barrier created by the cation-π and electrostatic interactions when $K^+$ ions interact with binding sites in the CPOS pores. This is further evidenced by the vibrations of characteristic groups, transport activation energy, and position of water molecules when ions sit in the cavity. Consequently, the enhanced $K^+$ transport rate, which is at least one order of magnitude higher than the $Na^+$ and $Li^+$ values, contributes to the ultrahigh selectivity. The high-performance synthetic $K^+$ ion nanochannel could provide an attractive strategy for developing artificial multifunctional porous materials with multiple length scales and spatial dimensions. We expect that the ultra-selective ion transport should prove valuable for bionanofluidic such as signal transmission, and electrochemical applications in solid-state energy storage devices. Furthermore, the artificial charged crystal pores open innovative horizons for understanding the role of cation-π and electrostatic interactions in ion transport within transmembrane channels in biological systems.

## Methods

**Characterizations.** Field-emission scanning electron microscopy (SEM, S-4800) with an accelerating voltage of 10 kV coupled with second electron (SE) imaging was employed to obtain further information on the structure and dimension. Powder X-ray diffraction (PXRD) measurements were accomplished on Micro Meritics Tristar II 3020 with a PANalytical B.V. Empyrean powder diffractometer using Cu-Kα radiation at 40 kV and 40 mA over a range of $2\theta = 4.0°$ up to 40.0° with a step size of 0.02° and 2 s per step. Fourier transform infrared (FTIR) spectra were measured using an Excalibur 3100 Fourier transform spectrometer (Varian, USA) to probe the detailed structures of the CPOS. The zeta potential of the CPOS was measured using a Zetasizer Nano ZS90 (Malvern Instruments). Thermogravimetric analysis (TGA) data were recorded by a TGA 500 thermogravimetric analyzer by heating with a rate of 10 °C min⁻¹ under the nitrogen atmosphere (60 ml min⁻¹). The $CO_2$ sorption experiments at 273 K up to 1 bar were collected using a Micromeritics ASAP 2020 surface area and pore size analyzer. Before the sorption analysis, the sample was evacuated at 150 °C for overnight using a turbo molecular vacuum pump. The micropore surface area was calculated from the $CO_2$ adsorption data by the Dubinin-Astakhov (DA) method. The pore size distributions were calculated according to the $CO_2$ adsorption isotherms using non-local density functional theory (NLDFT) method.

**Preparation of biomimetic $K^+$ channels by in situ growth strategy.** The single conical nanochannel were produced within a polyimide (PI) polymer membrane using the well-developed ionic track-etching technique. Prior to the chemical etching, the two sides of the PI membrane were respectively irradiated using an ultraviolet (UV) light for one hour. The single conical nanochannel was performed through the etching technique at 333 K. The PI membrane was clamped between two polytetrafluoroethylene (PTFE) compartments, of which one cell, facing the base of the conical nanochannel, was filled with 14% sodium hypochlorite as the etching solutions, while the other cell was filled with 1 M potassium iodide in order to neutralize the etchant as soon as the channel opened. Once finishing the etching, the residual solutions were removed thoroughly by soaking the membrane in MilliQ water (18.2 MΩ cm⁻¹). During the etching process, the carboxyl (−COOH) groups on the nanochannel surface emerged which is against the in situ growth of the CPOS material. To eliminate the disturbance, the amination was conducted using ethanediamine (Fig. 1b), resulting in the amino groups onto the carboxyl sites. In detail, the carboxyl surface was activated by soaking in an aqueous solutions of 150 mg 1-ethyl-3-(3-dimethylaminopropyl) carbodiimide (EDC) and 30 mg N-hydroxysuccinimide (NHSS) for one hour at room temperature without sunlight. At last, the samples were further soaked in 50 mM ethanediamine overnight. The amino PI nanochannel was obtained after washed for several times with MilliQ water.

In situ growth of the crystals within the single conical nanochannel by a bottom-up methodology in aqueous solutions under a static state without stirring to prevent any fluctuation during the crystal growth. For a typical reaction, 41.0 mg TBS powder was dissolved in 0.2 ml NaOH (1.0 M) solutions. Subsequently, 1.8 ml

$H_2O$ and 4.0 ml tetrahydrofuran (THF) were added into the solutions to form a mixture. Settled solutions (marked 1) were collected via a petcock from the mixture. Meanwhile, 23.0 mg DAB sample was dispersed into 2.0 ml $H_2O$ and 4.0 ml THF, then which were sonicated at 40 kHz frequency with the intervals of 10 s until the DAB sample was completely dissolved. Similarly, the settled solutions (marked 2) of the DAB mixture were also received. In order to make the crystals fill the single conical nanochannel, the tip of the single nanochannel faced the bottom of a reactor. The solutions marked 1 were first added into the reactor, which could bind onto the inside surface of the amino nanochannel. Finally, the solutions marked 2 were added, yielding in situ growth of the crystals from the tip to the base. The reactor was in a static state overnight. The PI membrane was washed for several times with $H_2O$/THF (1:2 of v:v) solutions. Note that SEM images show the intact and replete crystals within the base, indicating the successful growth from the tip to the base of the nanochannel.

**Current-voltage recordings.** The ionic transport properties through the unmodified and modified nanochannels were respectively measured by recording $I$-$V$ characteristics using a Keithley 6487. The single conical biomimetic nanochannel was mounted between two chambers of the PTFE cells. The two chambers were filled with the corresponding chloride salt solutions (LiCl, NaCl, KCl, $MgCl_2$, and $CaCl_2$) at the same concentration. Homemade Ag/AgCl electrodes in each chamber were used to apply an electric potential across the unmodified or modified membranes. Note that the anode faced the base of the nanochannel. The main transmembrane potential used in this work was a scanning voltage that varied from −1 to +1 V with a step bias of 0.1 V. All the recordings were carried out under the same conditions and each test was repeated for at least eleven times to obtain the average current values at different voltages. In addition, the whole tested system except the Keithley 6487 was located in a sealed chamber to avoid the interferences from $CO_2$ in air.

**Interaction between CPOS and ions by XPS measurements.** In order to confirm the strong interactions between the CPOS and metal ions, the CPOS materials were filled into the porous conical PI nanochannels (pore density: $10^7$ cm⁻²) using the in situ growth method, which underwent a $I$-$V$ scanning in different salt solutions (LiCl, NaCl, and KCl). Subsequently, the porous membrane (only used in the XPS experiments) was washed for several times with MilliQ water to remove possible residual ions from the surface of the CPOS materials. Dry membrane was investigated by XPS measurements to reflect the binding effects of the CPOS and metal ions.

**¹H NMR spectroscopy analysis.** NMR spectra were recorded on a Bruker DM300 or AV 400 spectrometer with tetramethylsilane (TMS) as the internal standard. Chemical shifts were quoted in parts per million (ppm) relative to the signals corresponding to the residual non-deuterated protons in NMR solvents. The ¹H NMR measurements of the R (5.5 mM), in the presence of various relevant metal ions ($Li^+$, $Na^+$ and $K^+$, 22 mM) in $D_2O$ solution, were performed to investigate the changes of the R related to ions ($D_2O$: δ 1.56 ppm).

**Conductance through CPOS pores.** The conductance ($G$) of the CPOS pores can be described as[21]:

$$G = nG_0 = \frac{I}{U} \tag{1}$$

where $I$ is the ion current obtained by current-voltage ($U$) recordings, $G_0$ is the conductance in a single CPOS pore and $n$ represents the amount of the CPOS pores in the PI nanochannel. Note that the used voltage is 0.1 V, where the $I$-$V$ curve exhibits an ohmic behavior. Besides, the temperature-dependent conductivity ($\kappa$) was measured by an intelligent thermoregulator (DC-2006) and A Keithley 6487 was used to monitor the $I$-$V$ characteristics at different temperatures. According to the Arrhenius equation (Supplementary Eq. 1), the relationship of the temperature and conductivity can be acquired. In this work, the dimension of the base and the tip of the conical nanochannel was measured by SEM images and equation, respectively. According to the multiple SEM images, we calculated the dimension of the base ($D_{base}$) to be ~750 nm. The dimension of the tip ($d_{tip}$) is calculated as follow (also see Supplementary Eq. 2):

$$d_{tip} = \frac{4IL}{\pi\kappa_c UD_{base}} \tag{2}$$

where $\kappa_c$ is the specific conductivity in 1 M KCl solution at 298 K, that is, 0.11173 Ω⁻¹ cm⁻¹. Therefore, the $d_{tip}$ was estimated to be ~75 nm. Conductivity (κ) can be calculated by the equation[51]:

$$\kappa = G\frac{4L}{\pi D_{base}d_{tip}} \tag{3}$$

where $L$ is the length of the PI nanochannel.

**Ion mobility through CPOS pores.** To further validate the transport properties of our biomimetic CPOS pores, the ion mobility was obtained by the typical drift-

diffusion experiments. Different from the $I-V$ characteristics, the drift-diffusion was performed using the applied voltages from $-0.2$ to $0.2$ V versus Ag/AgCl electrodes under a 10-fold concentration gradient. The sample was clamped between two PTFE compartments, of which one cell, facing the base of the conical nanochannel, was filled with 100 mM chloride salt solutions, while the other cell was filled with 10 mM same chloride salt solutions. We obtained the pure zero-current voltage ($E_m$) (Supplementary Table 3) according to the current-voltage tests. A redox potential generated on the electrodes should be deducted. Thus, the real $E_m$ (real $E_m = E_m - E_{redox}$) was subtracted from the redox potential ($E_{redox}$) which was calculated as follows[52]:

$$E_{redox} = \frac{RT}{zF} \ln \frac{\gamma_H c_H}{\gamma_L c_L} \qquad (4)$$

where $R$, $T$, $z$, and $F$ are the universal gas constant, temperature, ion valence, and Faraday constant, respectively. $\gamma$ stands for the activity coefficient of the high concentration (100 mM) and low concentration (10 mM), respectively. The real $E_m$ allows us to plot the mobility ratio, $\mu_+/\mu_-$, according to the Henderson equation[53]:

$$\frac{\mu_+}{\mu_-} = -\frac{z_+}{z_-} \frac{\ln\Delta - z_- FE_m/RT}{\ln\Delta - z_+ FE_m/RT} \qquad (5)$$

where $z$ refers to the valences of cations and anions, respectively. $F$, $R$, and $\Delta$ are the Faraday constant, universal gas constant, and ratio of concentration in the feed and permeation containers (here, $\Delta = 10$). $T = 298$ K, $z_+ = 1$, $z_- = -1$. We also measured the conductivity of various chloride solutions in high concentration ($c$) of 100 mM in order to neglect the surface charge contribution. The conductivity can then be described as:

$$\sigma = 10^3 (c_+\mu_+ + c_-\mu_-)N_A e \qquad (6)$$

where $c_+$ and $c_-$ are the concentrations of anions and cations, respectively. $N_A$ and $e$ are the Avogadro constant and the electron charge, respectively. Combining the Eq. (5) and Eq. (6), the cation mobility through the CPOS pores can be obtained and their values are also plotted in Fig. 4i, which can also be described as:

$$\mu_+ = \frac{10^{-3}\sigma(\ln 10 + FE_m/RT)}{2\ln 10 N_A ec} \qquad (7)$$

here, $\mu_+$ is only as the function of $\sigma$ and $E_m$.

**Ion transference number**. The property of cations through the negatively charged CPOS pores could be described by the cation transference number, which is given as:

$$t_+ = \frac{1}{2}\left(\frac{E_m}{\frac{RT}{zF}\ln\frac{\gamma_H c_H}{\gamma_L c_L}} + 1\right) \qquad (8)$$

where the $E_m$ refers to the real value and all the parameters could be found in Supplementary Table 3.

**Trinary ion permeation experiments**. The biomimetic nanochannel membrane was clamped between two PTFE compartments, of which one cell, facing the base of the conical nanochannel, was filled with a mixture of 100 mM LiCl, NaCl, and KCl as the feed solutions (10 ml), while the other cell was filled with MilliQ water as the permeation solution. The trinary ion selectivity measurements were conducted by applying a constant potential of $-1$ V versus Pt electrodes (Supplementary Fig. 13) using a Keithley 6487 picoammeter for 24 h. Once these measurements completed, the permeation solutions were collected and then measured using the inductively coupled plasma mass spectrometry for further information on metal ion concentration.

**Computational calculations**. To study the in-depth information on the hydration ions siting in the CPOS pores (that is, the crystal cavity), we carried out the Molecular Dynamics (MD) and Density Functional Theory (DFT) calculations. The crystal model and parameters could be adopted from our previous report[28]. In the presence of $K^+$, $Na^+$ and $Li^+$ ions, the system was a supercell composed of $2 \times 2 \times 3$-unit cells of the CPOS material, in which the CVFF force field parameters could be chosen to apply. The Verlet algorithm was employed to permit the numerical integration of motion equation. The equilibrium MD simulation was carried out according to the common NVT ensemble. The time step of 1 fs was adopted. For the crystal structure optimization, the kinetic time of 5 ps and the simulated temperature of 300 K were conducted. In the process of calculating structural relaxation, the kinetic time was set to $10^4$ fs. All the structures from the calculation optimizations were then thermally equilibrated at 300 K using the Berendsen thermostatic tank method. To investigate the electron density status of three types of ions in the CPOS cavity, we ran the DFT calculations in which the density distribution of hydrogen and oxygen were acquired by the Quantum Espresso software. The gradient corrected exchange-correlation functional calculations were carried out using the Generalized Gradient Approximation (GGA) method of the Perdew, Burke and Ernzerhof (PBE) function, with plane-wave basis sets imposing cutoff energies of 100 Ry. The convergence accuracy of the iterative

process (that is, self-consistent field, SCF) is $10^{-6}$. All simulations were run using the Large-scale Atomic/Molecular Massively Parallel Simulator (LAMMPS) code.

## Data availability
The data that supports the findings of this study can be found in the manuscript, its supplementary information, or are available from the corresponding author upon request.

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

## Acknowledgements

This work was supported by the National Key R&D Program of China (2017YFA0206904, 2017YFA0206900, 2021YFA1200400), the National Natural Science Foundation (21625303, 21905287, 21988102, 91956108, 21871103), and the Natural Science Foundation of Zhejiang Province (LZ22B010001). The membranes employed in this work are based on an UMAT experiment, performed at the beam line X0 at the GSI Helmholtzzentrum für Schwerionenforschung, Darmstadt (Germany) in the frame of FAIR Phase 0.

## Author contributions

L.W. conceived the research direction and guided the project. L.W. and W.X. designed the detailed project scope. W.X. designed and conducted the electrochemical tests and ion transport experiments. J.F. performed the preparation of the crystal material into the single nanochannel and provided the corresponding characteristics. L.W., W.X., T.B. and L.J. analyzed and discussed the experimental results and drafted the manuscript. W.X. carried out the numerical simulations. Y.Q. helped analyze the NMR data and FTIR results. J.F., L.F and X.-Y. K. joined the discussion of data and offered useful suggestions. L.W., T.B. and L.J. supervised the work.

## Competing interests

All authors declare no competing interests.
