## [Peer Review File · Nature Communications]

Biomimetic KcsA channels with ultra-selective K⁺ transport for monovalent ion sievingREVIEWER COMMENTS

Reviewer #1 (Remarks to the Author):

Review report for “Biomimetic KcsA channels with ultra-selective K⁺ transport for monovalent ion sieving”

The work “Biomimetic KcsA channels with ultra-selective K⁺ transport for monovalent ion sieving” addresses the transport behavior of conical nanochannels modified by a porous crystalline porous organic salt, CPOS (in this case, an organosulfonate–amidinium salt). The subnanometer pores of CPOS enable ultra-selective transport of K⁺ over similar monovalent ions, namely Li⁺ and Na⁺. The membrane, material and transport mechanism were extensively characterized by a variety of techniques and it was concluded that ultrasensitive K⁺ transport results from its lower hydration energy compared with the other cations and its interactions with the components of the CPOS salt.

A similar system was recently reported by other authors (Ref 21 in the manuscript) using a MOF instead of a CPOS. There is another recent related example using MOF (Guo, Y., Ying, Y., Mao, Y., Peng, X., & Chen, B. (2016). Polystyrene sulfonate threaded through a metal–organic framework membrane for fast and selective lithium-ion separation. *Angewandte Chemie*, 128(48), 15344-15348), which was not included in the references. That example reported very high Li⁺/Na⁺ and Li⁺/K⁺ selectivities, although did not include measurements at the single-pore level. The present work is novel in the sense that it reports high K⁺/Na⁺ and K⁺/Li⁺ selectivity, high K⁺ transport rates, uses a CPOS instead of a MOF, and has extensive characterization at the single-channel level.

The manuscript does not provide a strong point of the usefulness of having nanochannels with a very high K⁺ selectivity. For example, membranes with high Li⁺/Na⁺ selectivities can revolutionize Li⁺ extraction from brines, but I do not see a similar groundbreaking application for membranes with high K⁺ selectivity. Nevertheless, the results presented in this work are relevant from the point of view of rational material design and it is a very nice example of the integration of porous crystalline materials and nano/microporous membranes.

In many places, I find the paper rather confusing. This makes it difficult to scientifically evaluate the validity of the conclusions. There is also some lack of detail in the description of key experiments.

I ask the authors to please clarify/correct the following issues:

1) The use of “channel” and “nanochannel” is confusing, because in some places they describe the channel in the polymer (polyimide) membrane, but in others they refer to the pore of the CPOS. The authors should define in the introduction two different terms for these two different types of channels and stick to them along the paper. I think it is correct to reserve “nanochannel” for the channel in the polymer membrane, and probably use “pore” for the pores in the CPOS. Again, defining the meaning of these words at the introduction will help a lot.

2) In page 5, the authors use the formula:

$$G = 10^3 * (\mu_+ + \mu_-) * c_b * N_A * e * w * h / l + 2 * \mu_+ * \sigma * w / l$$

to model the conductivity of CPOS modified membrane. I have reread this section a couple

of times and I still do not get if the channel of width w and length l that this formula refers to is the nanochannel in the polymer membrane filled with CPOS or the sub-nm pores of the CPOS. In both cases, the use of the formula is incorrect.

The formula has two terms, which model bulk conduction (first term) and surface conduction (second term). If the authors are using the formula to model the conical nanochannel in the polymer membrane filled with CPOS, then its use is incorrect because the channel is filled with the CPOS, which is a permselective material (so, only cation conductivity should be included in the first term). Moreover, the walls of the nanochannel in the polymer membrane are coated by NH_2 (see Figure 1b, middle panel), so the surface should conduct anions, not cations. Finally, the formula is valid for a cylindrical channel (otherwise, does w refer to the tip or to the base?).

If the authors are using the formula to model the sub-nm pores in CPOS, then it is also incorrect. The formula is valid for pores wider than the Debye length of the solution (and much much wider than the size of the ions). Modelling a sub-nm pore as having distinctly bulk and a surface regions is incorrect because the size of the ions is similar to the size of the channel.

How was G obtained? From the slope around $V = 0$?

Ref 43 is also unappropriated for the proposed equation because that work from Cervera and coworkers refers to a conical channel, while the above equation is valid for cylindrical channels.

Last but not least, the authors use the formula to draw a line in Figure 2c, but they do not analyze if the slope of the line is consistent to what it is expected from the equation above.

In summary, I find the use of this model by far the weakest and most confusing part of the paper. This model should either be used and explained correctly or completely dropped, so it won't confuse the readership.

3) If I understand correctly, some experiments (e.g. I-V curves in Figure 2) were done with a membrane containing a single nanochannel filled with CPOS, where others (XPS measurements) involved a membrane with multiple nanochannels. Is this correct? What's about permeation experiments? The authors should specify which type of membrane was used in each type of experiment, for all experiments in the paper. If a multiple nanochannel membrane was used, the surface density of nanochannels (nanochannels/ cm^2) should be state. Which were the volumes of the solution in the permeation experiments?

4) Figure 4a, in order to draw conclusions from the $g(r)$ of the ions inside CPOS, the $g(r)$ of the same ions in bulk water calculated with the same force-field should be provided for comparison.

5) Methods section, "Preparation of biomimetic K^+ channels by in situ growth strategy", this section does not clarify how the single (or multiple) ion-tracks in the polymer membrane were generated before etching. Researchers unfamiliar with this technology will find the description of the authors very confusing. Related to this question, what is a "UMAT experiment"

6) The authors claim that K^+ interacts strongly than other ions with the pore walls, and that

results in its higher transport rate:

"Taken together, our experiments and MD simulations indicate that the high K⁺ transport and selectivity arise from the low transport barrier created by the cation- π and electrostatic interactions when K⁺ ions interact with binding sites in the CPOS channels"

The simulation results suggest that K⁺ is strongly dehydrated than the other ions, supporting this claim (although comparison with bulk $g(r)$'s is still necessary, see my comment above). However, NMR results in Figure 3b show that the largest shifts from the pristine material occurs for Li⁺, not for K⁺, wouldn't this contradict the author hypothesis? Also, the NMR spectra in Figure 2d show qualitative changes after the addition of the ions, namely there are less peaks in the spectra of the ion-treated material than in the spectrum of the pristine one. What is the origin of this change?

7) Supplementary Figure 12: There are clear oscillations in the current vs time plots. What is their origin? (electronic noise?).

8) In SEM experiments, was a single-channel membrane used? If so, how do the authors find the single channel in the membrane for SEM observation?

There are some additional experiments/discussions that can really enhance this work. I encourage the authors to consider them:

9) The selectivity of the channel is K⁺ > Na⁺ > Li⁺. What happens with Cs⁺ and Rb⁺ ? Do they follow the trend or does the selectivity experience a maximum with respect to the ionic radii?

10) Does Figure 1b (last panel) imply that there is some orientation of the CPOS crystal within the nanochannel? Is a preferential orientation required for the transport behaviors proposed by the authors?

Finally, a few minor issues:

11) The title of Figure 2 "Selective ion transport through the biometric K⁺ channels." is misleading. Selectivity in transport rate is shown in Figure 4. Figure 2 is focus in current rectification (which also shows dependence on the type of ion, but the focus of the figure is rectification, not transport selectivity).

12) Page 2, 10⁴ magnitudes = four orders of magnitude?

13) Page 5, line 163, I think the authors wanted to say "remarkable" instead of "markable"

14) Page 5, line 164, Fig 2b is not a histogram (does not show frequency in the y axis), it is a bar plot.

Reviewer #2 (Remarks to the Author):

In this manuscript, Ben and Wen report construction of a highly selective K⁺ channel by the engineered nanopore, which are filled with in-situ grown nanoporous crystals. The selectivity for K⁺ over the other monovalent cations like Li⁺ and Na⁺ is at the highest level among the reported nanopores so far, and also the transport rate is quite high. The authors carefully conducted characterization of the

materials and investigated the transporting features by both experimental and theoretical analyses. Together with the theoretical simulation, the mechanism of the high selectivity could be rationally explained. Thus results and discussions described in the manuscript would have certain impacts on the researchers in the relevant fields. On the other hand, there are still some issues that need to be addressed appropriately.

1. The origin of the ion selectivity was investigated based on the interactions of the ions and water molecules in the nanochannel formed in the CPOS micro crystals. While this approach seems reasonable, the microcrystals grown in the nanochannel should have grain boundaries, which would allow ions to pass on the surface of the microcrystals and would crucially influence on both the selectivity and conductivity of ions. It is important to clarify whether the condition of the crystal growth such as concentration of the material, temperature, solvent, would influence on the quality and size of the microcrystals, and also their properties like selectivity and transport rate of ions would be changed by these factors.
2. It is difficult to understand the necessity of the conical morphology of the nanopore for the high selectivity of the current system, since the origin of the selectivity was explained based on the structure of the nanochannel formed in the CPOS microcrystal and the hydration/dehydration of the transported ions. Without proving the importance of the conical shape of the nanochannel, it is not suitable to claim the similarity of the current design with KcsA channel.
3. The list of precedent systems summarized in Supplementary Table 2 is quite informative for the researchers in relevant community and would be appreciated. Based on this, the authors emphasized that the current system revealed the highest K^{+} selectivity over Li^{+} and Na^{+} compared with other precedent systems. However, the nanopores reported in ref #9 had a higher Na^{+}/K^{+} selectivity than the current system. In addition it is simply true that Li^{+}/K^{+} and Li^{+}/Na^{+} were not reported in #9, and this doesn't mean that the authors' system is superior to that reported in #9.
4. The authors claimed that the high K^{+} selectivity was originated from the low hydration energy of K^{+} compared with other cations. If so, is the mobility of Rb^{+} is higher than K^{+} ?
5. Why did K^{+} show the highest activation energy for the ion transport among Li^{+} , Na^{+} and K^{+} , although it reveals the lowest hydration energy?
6. Why did the divalent cations show the ion flux at the positive voltages while the monovalent ones did not as shown in Supplementary Figure 7a?
7. CPK model would be helpful for understanding the size of the nanochannel shown in Fig. 1 (iii).

Reviewer #3 (Remarks to the Author):

The manuscript submitted by Li et al. presents a strategy to construct porous subnanometer structures displaying selective K^{+} transport. This strategy relied on the use of a covalent organic framework hosted in an asymmetric nanochannel.

The set of experiments is interesting, but I consider that this work lacks novelty. The use of metal-organic frameworks and covalent organic frameworks hosted in nanochannels for controlling ion transport has been previously reported by the same research group:

- Fast and selective fluoride ion conduction in sub-1-nanometer metal-organic framework channels, Li et al. Nat. Comm. 2019.

- Unidirectional and Selective Proton Transport in Artificial Heterostructured Nanochannels with Nano-to-Subnano Confined Water Clusters, Li et al. Adv. Mater, 2020.

- Ultraselective Monovalent Metal Ion Conduction in a Three-Dimensional Sub-1 nm Nanofluidic Device Constructed by Metal–Organic Frameworks, Lu et al. ACS Nano, 2021.

This work describes the exploration of a covalent organic framework that has not been previously reported in prior studies; however, no real advance is reported. For example, one key parameter of these hybrid membranes is the rectification efficiency. Values reported in this manuscript are close to 20 at best (Fig. 2b). This value is well below rectification efficiency values reported almost 15 years ago (Vlassiuk and Siwy, Nano Letters, 2007).

Given the lack of novelty, I do not recommend the publication of this paper in Nature Communications.

Response to the comments of the reviewers

We appreciate the constructive comments and suggestions from the reviewers, and we have revised the manuscript accordingly as detailed in the responses below. The corresponding changes have been highlighted in yellow in the main text and supplementary information.

Reviewer #1 (Remarks to the Author):

The work “Biomimetic KcsA channels with ultra-selective K⁺ transport for monovalent ion sieving” addresses the transport behavior of conical nanochannels modified by a porous crystalline porous organic salt, CPOS (in this case, an organosulfonate–amidinium salt). The subnanometer pores of CPOS enable ultra-selective transport of K⁺ over similar monovalent ions, namely Li⁺ and Na⁺. The membrane, material and transport mechanism were extensively characterized by a variety of techniques and it was concluded that ultrasensitive K⁺ transport results from its lower hydration energy compared with the other cations and its interactions with the components of the CPOS salt.

A similar system was recently reported by other authors (Ref 21 in the manuscript) using a MOF instead of a CPOS. There is another recent related example using MOF (Guo, Y., Ying, Y., Mao, Y., Peng, X., & Chen, B. (2016). Polystyrene sulfonate threaded through a metal–organic framework membrane for fast and selective lithium - ion separation. *Angewandte Chemie*, 128(48), 15344-15348), which was not included in the references. That example reported very high Li⁺/Na⁺ and Li⁺/K⁺ selectivities, although did not include measurements at the single-pore level. The present work is novel in the sense that it reports high K⁺/Na⁺ and K⁺/Li⁺ selectivity, high K⁺ transport rates, uses a CPOS instead of a MOF, and has extensive characterization at the single-channel level.

The manuscript does not provide a strong point of the usefulness of having nanochannels with a very high K⁺ selectivity. For example, membranes with high Li⁺/Na⁺ selectivities can revolutionize Li⁺ extraction from brines, but I do not see a similar groundbreaking application for membranes with high K⁺ selectivity. Nevertheless, the results presented in this work are relevant from the point of view of rational material design and it is a very nice example of the integration of porous crystalline materials and nano/microporous membranes.

In many places, I find the paper rather confusing. This makes it difficult to scientifically evaluate the validity of the conclusions. There is also some lack of detail in the description of key experiments.

Response: We appreciate the reviewer for the positive comments.

Firstly, different from COF and MOF materials, self-assembled crystalline porous organic salts (CPOSSs) with one-dimensional polar pores containing water molecules, which play an important role in ion transport of the materials. In this work, we demonstrate that the nanoconfined CPOS has not only well-defined crystalline structures but also the high surface charge for selective cation transport. Recently, filling function materials into a nanoconfined space to obtain a selective ionic rectifier has been considered to be key for significant physiological events in neurons or cardiac cells (*Nat. Mater.* 2020, 19, 701-702). This method was also employed in our work to investigate the properties of different ions across CPOS nanochannels. We found that the CPOS-filled nanochannels achieve ultrahigh K⁺ transport, showing great potential in separation and sensing applications, such as water desalination and metal ion trace detection.

Secondly, we thank the reviewer for the reminder. The paper (*Angew. Chem. Int. Ed.* 2016, 128,

15344-15348) helps us better understand the selective ion transport in nanoconfinement, which has been included in the references (as ref. 24).

Thirdly, thanks again for your valuable suggestions. Besides separating metal ions, the membrane with high-efficient potassium ion transport demonstrates the construction of artificial potassium channels to simulate and study the behavior of life in vivo such as neural signal transmission and ion channelopathy. Owing to the attractive properties of biological ion channels, especially their high selectivity and ultrafast transport ability, researchers are trying to obtain the membrane with bioinspired ion channels for possible applications. Here, we successfully prepare a composite membrane with ultrahigh potassium ion selectivity. Besides, membranes with ultrahigh potassium ion transport can be also used in high-rate solid-state energy storage devices. We have added related discussions in the Introduction and Conclusion sections of the revised manuscript. We also provide point-by-point responses as follows.

I ask the authors to please clarify/correct the following issues:

Comment 1: The use of “channel” and “nanochannel” is confusing, because in some places they describe the channel in the polymer (polyimide) membrane, but in others they refer to the pore of the CPOS. The authors should define in the introduction two different terms for these two different types of channels and stick to them along the paper. I think it is correct to reserve “nanochannel” for the channel in the polymer membrane, and probably use “pore” for the pores in the CPOS. Again, defining the meaning of these words at the introduction will help a lot.

Response: Thanks very much for the reviewer’s suggestions. We added the definition of the “nanochannel” for the channel in the polymer membrane, and “pore” for the pores in the CPOS in the introduction of the revised manuscript. And we use the corrected terms throughout the whole revised manuscript.

Comment 2: In page 5, the authors use the formula:

$$G=10^3(\mu_+ + \mu_-)c_b N_A e w h / l + 2\mu_+ \sigma w / l$$

to model the conductivity of CPOS modified membrane. I have reread this section a couple of times and I still do not get if the channel of width w and length l that this formula refers to is the nanochannel in the polymer membrane filled with CPOS or the sub-nm pores of the CPOS. In both cases, the use of the formula is incorrect.

The formula has two terms, which model bulk conduction (first term) and surface conduction (second term). If the authors are using the formula to model the conical nanochannel in the polymer membrane filled with CPOS, then its use is incorrect because the channel is filled with the CPOS, which is a permselective material (so, only cation conductivity should be included in the first term). Moreover, the walls of the nanochannel in the polymer membrane are coated by NH_2 (see Figure 1b, middle panel), so the surface should conduct anions, not cations. Finally, the formula is valid for a cylindrical channel (otherwise, does w refer to the tip or to the base?).

If the authors are using the formula to model the sub-nm pores in CPOS, then it is also incorrect. The formula is valid for pores wider than the Debye length of the solution (and much much wider than the size of the ions). Modelling a sub-nm pore as having distinctly bulk and a surface regions is incorrect because the size of the ions is similar to the size of the channel.

How was G obtained? From the slope around $V = 0$?

Ref 43 is also unappropriated for the proposed equation because that work from Cervera and

coworkers refers to a conical channel, while the above equation is valid for cylindrical channels. Last but not least, the authors use the formula to draw a line in Figure 2c, but they do not analyze if the slope of the line is consistent to what it is expected from the equation above. In summary, I find the use of this model by far the weakest and most confusing part of the paper. This model should either be used and explained correctly or completely dropped, so it won't confuse the readership.

Response: We appreciate the reviewer very much for the comments and detailed explanations. In this experiment, we obtained the ionic conductance of the functional single-nanochannel by recording the current-voltage curves, which can be defined as

$$G = \frac{I}{U} \times \frac{l}{S}$$

where I is the ion current obtained by current-voltage (U) recordings. l is the length of the PI nanochannel and S is the cross-sectional area. Considering the effective ionic flux through the CPOS material and the base facing the anode, the S can be roughly calculated based on the dimension of the base (D_{base}). For the current-voltage values, we selected the current values at the same applied voltage to calculate the corresponding results according to the previously reported papers (*Nat. Mater.* 2020, 19, 767-774; *ACS Appl. Mater. Interfaces* 2019, 11, 35496-35500; *Nat. Nanotechnol.* 2020, 15, 307-312.). The details are also added in the "Conductivity through CPOS pores" section of the revised manuscript.

For the formula of $G=10^3(\mu_+ + \mu_-)c_b N_A ewh/l + 2\mu_+ \sigma w/l$, the first part of the equation corresponds to the Ohmic conductance due to the bulk concentration of ions (c_b), and the second part is the contribution from the excess counterions; the width (w) and length (l) refer to the width and length of a single CPOS pore, as shown in the below diagram (Figure R1a). The CPOS pore is simplified as a cylindrical pore. Using this formula to explain surface-charge-governed ion transport in subnanometer channels can also be found in some works (*J. Am. Chem. Soc.* 2012, 134, 16528–16531; *J. Am. Chem. Soc.* 2017, 139, 6314–6320; *Proc. Natl. Acad. Sci. USA* 2020, 117, 13959-13966; *Nano Lett.* 2017, 17, 728-732; *Phys. Rev. Lett.* 2009, 102, 256804; *Nano Lett.* 2006, 6, 89–95). In these works, the width ranges from ~0.5 nm to 10.4 nm, and the width of the CPOS pore in our work is 1.09 nm which is within this range.

The line in Figure 2c was drawn by fitting the bulk values that is the conductivity when the solution concentrations are 3, 2, and 1 M. The perfect fitting result shows that the ion transport behavior is consistent with the bulk phase behavior in high-concentration solution, and also agrees well with the first part of the formula. Besides, in these solutions, the width of the COPS pores (1.09 nm) is much larger than the Debye length (0.3 nm) according to the Debye-Hückel equation (*Sci. Adv.* 2021, 7, eabe9924; *Angew. Chem. Int. Ed.* 2020, 59, 8720-8726; *PNAS* 2020, 117, 13959-13966). When the concentration is ≤ 0.1 M, the ionic conductance becomes independent of the ionic concentrations and is dominated by the surface-charge-governed ions transport, in which the thickness of electric double layer increases and becomes comparable with the dimension of the pore, as shown in Figure R1b (*Nano Lett.* 2017, 17, 728–732).

Figure R1. (a) Schematic of the simplified model of a single-pore CPOS. (b) The Debye length as a function of the ion concentration. The Debye length in 10^{-1} M approximates the width of CPOS pore; therefore, when the ion concentration is below 10^{-1} M, the Debye length overlaps and ion transport is surface-charge-governed; when the ion concentration is greater than 10^{-1} M, the Debye length is lower than the width of CPOS pore and thus, the ion transport shows bulk behavior, less unaffected by the surface charge.

In addition, the formula of $G=10^3(\mu_+ + \mu_-)c_b N_A e w h / l + 2\mu_+ \sigma w / l$ was employed to evaluate the conductance of a single nanofluidic channel/pore (with a sub-1 nm size and cation selectivity) filled with electrolyte solution (*J. Am. Chem. Soc.* 2012, 134, 16528–16531; *J. Am. Chem. Soc.* 2017, 139, 6314–6320; *PNAS* 2020, 117, 13959–13966). The ionic conductance exhibits a close relationship with the concentration of the electrolyte. The measured conductance gradually deviates from bulk value in low-concentration solution. In general, the conductance of a nanofluidic channel/pore filled with electrolyte solution is composed of two parts: bulk conductance contributed by the bulk electrolyte and surface conductance contributed by the surface charge which can be qualitatively described by using the equation. We are very grateful to the reviewer for giving us sufficient explanation on the use of the formula, which help us better understand its essence. We have rewritten the part in the revised

manuscript as follow.

“The affinity of the concentration (c_b) and conductivity (G) in a single CPOS pore can be qualitatively described as $G=10^3(\mu_++\mu_-)c_bN_Aewh/l+2\mu_+\sigma w/l$ (ref. ^{43,44}), where μ , N_A , and e represent the ion mobility, Avogadro constant, and elementary charge, respectively; w and l are the width and length of a single CPOS pore, respectively; h is the height of a CPOS cavity, and σ is the surface charge density. In high-concentration region, the ion conductivity is linearly proportional to c_b with the $R^2 = 0.995$ that is in good agreement with the bulk term of $10^3(\mu_++\mu_-)c_bN_Aewd/l$. Moreover, the conductivity in low-concentration region, also defined as $2\mu_+\sigma w/l$, comes from the contribution of the ions accumulated in strong Debye layer overlaps. In that case, a conductivity plateau insensitive to the solution concentration occurs in which the surface charge dominates ion transport.”

Comment 3: If I understand correctly, some experiments (e.g. I-V curves in Figure 2) were done with a membrane containing a single nanochannel filled with CPOS, where others (XPS measurements) involved a membrane with multiple nanochannels. Is this correct? What’s about permeation experiments? The authors should specify which type of membrane was used in each type of experiment, for all experiments in the paper. If a multiple nanochannel membrane was used, the surface density of nanochannels (nanochannels/cm²) should be state. Which were the volumes of the solution in the permeation experiments?

Response: We appreciate the reviewer very much for the comments. We performed the XPS measurements on the membrane with multiple nanochannels (surface density of multiple nanochannels: 10⁷ nanochannels/cm²) for successful observation of the CPOS materials, which was specified in the “*Interaction between CPOS and ions by XPS measurements*” in Methods. We have also specified the type of membrane used in each type of experiment in the Method section. The volume of the solution in the permeation experiments is 10 ml. These details have been added in the revised manuscript.

Comment 4: Figure 4a, in order to draw conclusions from the $g(r)$ of the ions inside CPOS, the $g(r)$ of the same ions in bulk water calculated with the same force-field should be provided for comparison.

Response: We appreciate the reviewer very much for the suggestions. We have supplemented the data of the $g(r)$ of the same ions in bulk water calculated with the same force-field, as shown in Figure R2. We can find that the distance of oxygen from ion in bulk water (Figure R2a) becomes longer than that in CPOS pores (Figure R2c) due to the interaction between oxygen atoms and the walls of CPOS pores. Similarly, the hydrogen atoms are more closed to the metal ion due to the rotation of hydrated water molecular in nanoconfined CPOS pores (Figure R2c, d). These calculated data are consistent with the experimental results. We also added these data into the Supplementary Information.

Figure R2. RDFs of the ion–oxygen (a) and ion–hydrogen (b) distances in bulk water calculated using the same force-field with that in CPOS pores (c, d).

Comment 5: Methods section, “Preparation of biomimetic K^+ channels by in situ growth strategy”, this section does not clarify how the single (or multiple) ion-tracks in the polymer membrane were generated before etching. Researchers unfamiliar with this technology will find the description of the authors very confusing. Related to this question, what is a “UMAT experiment”

Response: We appreciate the reviewer very much for the comments. In fact, the ion-track membranes were purchased from GSI Helmholtzzentrum für Schwerionenforschung, Germany. Before etching, these membranes were treated by specific energy (11.4 MeV/n, Ion: Au) in Germany. The “UMAT experiment” is an experiment ID from GSI. Since we use these as-prepared membranes, the statement of “*The membranes employed in this work are based on an UMAT experiment, performed at the beam line X0 at the GSI Helmholtzzentrum für Schwerionenforschung, Darmstadt (Germany) in the frame of FAIR Phase 0*” is required to be given in our manuscript.

Comment 6: The authors claim that K^+ interacts strongly than other ions with the pore walls, and that results in its higher transport rate:

“Taken together, our experiments and MD simulations indicate that the high K^+ transport and selectivity arise from the low transport barrier created by the cation- π and electrostatic interactions when K^+ ions interact with binding sites in the CPOS channels”

The simulation results suggest that K^+ is strongly dehydrated than the other ions, supporting this claim (although comparison with bulk $g(r)$'s is still necessary, see my comment above). However, NMR results in Figure 3b show that the largest shifts from the pristine material occurs for Li^+ , not for K^+ , wouldn't this contradict the author hypothesis? Also, the NMR spectra in Figure 2d show qualitative changes after the addition of the ions, namely there are less peaks in the spectra of the ion-treated material than in the spectrum of the pristine one. What is the origin of this change?

Response: We appreciate the reviewer very much for the comments. We propose that in biomimetic analogs, ions are first adsorbed onto the pore surface and lose most of the outer hydration shells, which is necessary to overcome the hydration energy barrier required to enter their nanochannels.

NMR results show the largest shifts for Li^+ which implies that the Li^+ should have the highest energy barrier, impeding Li^+ transport through CPOS pores. On the contrary, NMR spectra shows that K^+ could more easily pass the CPOS pores. These changes of the NMR peaks stem from the interaction between ions and surface of the CPOS, which includes the cation- π and electrostatic interactions. We propose a possible reason that these shifts observed with respect to the reference peak may be attributed to the shielding effect of the coordinating donor atom electron density of compounds around the metal ions. For the simulation results, the oxygen distribution in the cation-entered CPOS indicates that the oxygen atoms accumulate inside the hydration shells of the cations (Figure R3a). According to these simulations (Figure R3b), a higher oxygen density around the Li^+ ions suggested that it is difficult for the Li^+ ions to shed the surrounding water molecules (hydrated Li^+ ion is confined in the pores). The oxygen density around K^+ ions is lower and its RDF (Oxygen) distribution is wider, implying that K^+ ions can more easily remove water molecules (dehydrated K^+ ion could easily move) in the pores for fast transport. In the nanoconfined space, especially in the angstrom scale, the water molecules surrounding the ion tend to strongly interact with the surface of the pore walls (*Nat. Mater.* 2016, 15, 850-855.). It can be seen that the hydration layer of Li^+ ion is like the hard hydration shell, which is easy to be trapped in the CPOS pores rather than contributing fast transport. The hydration layer of K^+ ion is softer and easier to dehydrate and rapidly transport in the CPOS pores (*Nat. Commun.* 2019, 10, 850-856.), and these results are also consistent with the NMR measurements. We would like to thank the reviewers again for their professional comments, which help us to supplement a more comprehensive discussions for these results. These contents have been added into the revised manuscript.

Figure R3. (a) Theoretical RDFs of steady-state ion–oxygen and ion–hydrogen distances in the CPOS pores. (b) Hydrogen and oxygen atomic density profiles along the z–axis for Li^+ , Na^+ , and K^+ around the screwing cavity.

Comment 7: Supplementary Figure 12: There are clear oscillations in the current vs time plots. What is their origin? (electronic noise?).

Response: We appreciate the reviewer very much for the comments. As shown in Figure R4a, the ionic current at -1 V is far higher than that at $+1$ V through the CPOS pores. In Figure R4b, we can find that the ionic current is the nA level (the original current level in Supplementary Figure 12 is pA that was a typo and now was corrected), which is close to the detection limit of this instrument. Such a variation is common in I-T curves. In Figure 4c, we could find that the ionic current become stable

without clear oscillations at -1 V bias due to larger current values which is far higher than the detection limit.

Figure R4. (a) I-V curve of the functional nanochannel. (a, b) Ionic current under a constant bias. For comparison, two constant biases of $+1$ V (a) and -1 V (b) were respectively applied to confirm the effective voltage control for the high-performance ion selectivity.

Comment 8: In SEM experiments, was a single-channel membrane used? If so, how do the authors find the single channel in the membrane for SEM observation?

Response: We appreciate the reviewer very much for the comments. In PI membrane, the conical nanochannel has a base with a ~ 750 nm diameter and a tip with a ~ 75 nm diameter. The base could be found as long as we seek carefully. However, the tip is too small to be found in such a large membrane. Therefore, we used a porous membrane (10^7 nanochannels/cm²) obtained by appropriate etching for SEM observation, reflecting the microscopic morphology of the tip region. These details have also been added into *Supplementary Materials and Methods*.

There are some additional experiments/discussions that can really enhance this work. I encourage the authors to consider them:

Comment 9: The selectivity of the channel is $K^+ > Na^+ > Li^+$. What happens with Cs^+ and Rb^+ ? Do they follow the trend or does the selectivity experience a maximum with respect to the ionic radii?

Response: We appreciate the reviewer very much for the significant suggestions. We performed the ion permeation experiments to show the selectivity of Cs^+ and Rb^+ in a mixture (100 mM LiCl, NaCl, KCl, RbCl, and CsCl as the feed solutions) through the CPOS pores. The other cell was filled with MilliQ water as the permeation solution. The ion concentration in the permeation was obtained by the inductively coupled plasma mass spectrometry (ICP-MS) measurements. As shown in Figure R5a, the CPOS pores have a selective permeation with a rank of $K^+ \gg Na^+ > Li^+ > Rb^+ > Cs^+$, which show the ultra-selective K^+ transport. According these results, the permeation of Rb^+ is larger than that of Cs^+ because Rb^+ has a smaller ion radius and radius of the first hydration shell (see Table R1). In this case, the Rb^+ could more easily enter and pass the CPOS pores. Although the hydration energy of Cs^+ is lower than that of Rb^+ , the size effect might play a more important role in the selective transport. Based on the permeation results, the ion selectivity of K^+/Cs^+ and K^+/Rb^+ is up to ~ 1300 and ~ 110 , which is an excellent advance in the monovalent ion sieving (Figure R5b). Additionally, the ion selectivity of K^+/Li^+ and K^+/Na^+ drops due to the competitive penetration in the quinary mixed solutions (*Sci. Adv.* 2020, 6, eabd9045; *Angew. Chem. Int. Ed.* 10.1002/anie.202108801). These results indicate that this CPOS materials could show potential in the removal of the radioactive element such as isotopes of cesium. These contents have been added in Supplementary Information to support our view of the wide application of the CPOS materials.

Figure R5. (a) Ion permeation through the CPOS pores under an external bias of -1 V . (b) Ion selectivity of the CPOS pores.

Table R1. Ion-dependent parameters including the ionic radius, hydration ionic radius, number of the first hydration shell, and hydration energy.

Ion type	Ionic radius (Å)	Hydration ionic radius (Å)	Radius (R_{\min}) of the first hydration shell (Å)	Hydration energy (kcal mol^{-1}) in kT
Li^+	0.60	3.82	2.08	122.2 ± 0.6
Na^+	0.95	3.58	2.36	98.8 ± 0.8
K^+	1.33	3.31	2.80	80.9 ± 1.0
Rb^+	1.48	3.29	2.89	75.5 ± 0.9
Cs^+	1.69	3.29	3.14	67.7 ± 0.7
Cl^-	1.81	3.32	2.24	75.8

Comment 10: Does Figure 1b (last panel) imply that there is some orientation of the CPOS crystal within the nanochannel? Is a preferential orientation required for the transport behaviors proposed by the authors?

Response: We appreciate the reviewer very much for the comments. The arrow in Figure 1b does not present the orientation of the CPOS crystal within the nanochannel, but shows that the CPOS crystal grows from the tip to the base of the PI nanochannel, which is beneficial to the filling of the crystal into the whole PI nanochannel. Once the growth direction is reversed, the crystals in the nanochannel cannot be completely filled especially in the tip region, leading to a vacancy that could produce a deviation of ion transport. In our work, the used functional nanochannels were filled in the

same way to maintain the consistence of the results. These discussions have been added into the Supplementary Figure 2 (Page 28).

Finally, a few minor issues:

Comment 11: The title of Figure 2 “Selective ion transport through the biometric K⁺ channels.” is misleading. Selectivity in transport rate is shown in Figure 4. Figure 2 is focus in current rectification (which also shows dependence on the type of ion, but the focus of the figure is rectification, not transport selectivity).

Response: Thanks for the reviewer’s comments. We have changed the title of Figure 2 to “Asymmetric ion transport behaviors of the biometric K⁺ channels”.

Comment 12: Page 2, 10⁴ magnitudes = four orders of magnitude?

Response: Thanks for the reviewer’s comments. We have corrected it by using the expression of “four orders of magnitude”.

Comment 13: Page 5, line 163, I think the authors wanted to say “remarkable” instead of “markable”

Response: Thanks for the reviewer’s comments. We have corrected the typo.

Comment 14: Page 5, line 164, Fig 2b is not a histogram (does not show frequency in the y axis), it is a bar plot.

Response: Thanks for the reviewer’s comments. We have changed the expression of this chart to a “bar plot”.

Reviewer #2 (Remarks to the Author):

In this manuscript, Ben and Wen report construction of a highly selective K⁺ channel by the engineered nanopore, which are filled with in-situ grown nanoporous crystals. The selectivity for K⁺ over the other monovalent cations like Li⁺ and Na⁺ is at the highest level among the reported nanopores so far, and also the transport rate is quite high. The authors carefully conducted characterization of the materials and investigated the transporting features by both experimental and theoretical analyses. Together with the theoretical simulation, the mechanism of the high selectivity could be rationally explained. Thus results and discussions described in the manuscript would have certain impacts on the researchers in the relevant fields. On the other hand, there are still some issues that need to be addressed appropriately.

Response: We appreciate the reviewer’s high comments and support of our work.

Comment 1: The origin of the ion selectivity was investigated based on the interactions of the ions and water molecules in the nanochannel formed in the CPOS micro crystals. While this approach seems reasonable, the microcrystals grown in the nanochannel should have grain boundaries, which would allow ions to pass on the surface of the microcrystals and would crucially influence on both the selectivity and conductivity of ions. It is important to clarify whether the condition of the crystal growth such as concentration of the material, temperature, solvent, would influence on the quality and size of the microcrystals, and also their properties like selectivity and transport rate of ions would be changed by these factors.

Response: We appreciate the reviewer very much for the suggestions. As you said, the conditions of the crystal growth such as concentration of the material, temperature, solvent indeed influence the quality and size of the microcrystals. Before this work, we investigated the impact of the synthesis conditions on the crystal growth. Then we compared the selectivity and transport rate of ions of different crystals and selected the optimal condition. As shown in Figure R6, CPOS (a, i), that is the product used in this work, shows the excellent crystal morphology with uniform size, which is beneficial to form compact filling in PI nanochannel. We used the concentration of the monomer materials such as TBS and DAB from 0.5 times [20.5 mg+11.5 mg (a, iii)] to 2 times [82.0 mg+46.0 mg (a, ii)]. It was found that the crystallinity (a, ii) is not as good as CPOS (a, i), which is originated to the uncontrollable nucleation and growth. When using the low concentration of the materials, the nucleation and growth were restrained and no crystal was found (a, iii). Similarly, high temperature accelerates the process of nucleation and growth, however, resulting in the poor crystal morphology (b, ii and iii). In addition, the polarity of solvents plays an important role in the crystal nucleation and growth. For example, we used three solvents including THF, CH₃OH, and H₂O with a polarity rank of THF < CH₃OH < H₂O. SEM images indicate that highly polar solvents (c, ii and iii) destroy the integrity of the crystals, forming small and unevenly crystal sizes, which might impede the growth of CPOS in PI nanochannel and the high-performance ion transport.

Figure R6. SEM images of CPOS obtained in different conditions. (a) Concentration of the materials such as [41.0 mg TBS+23.0 mg DAB] (i), [82.0 mg TBS+46.0 mg DAB] (ii), and [20.5 mg TBS+11.5 mg DAB] (iii). (b) Temperature such as 25 (i), 40 (ii), and 55°C (iii). (c) Solvent such as THF (i), H₂O (ii), and CH₃OH (iii).

In addition, the XRD measurements, as shown in Figure R7, reveal that all of the CPOS materials possess the same crystal structures, with characteristic peaks such as (200), (400), (4-20), (4-11), (440), (431), (6-20), (640), (721), (840), etc.

Figure R7. XRD patterns of CPOS obtained in [25°C THF 41.0 mg+23.0 mg] (a), [40°C THF 41.0 mg+23.0 mg] (b), [55°C THF 41.0 mg+23.0 mg] (c), [25°C THF 82.0 mg+46.0 mg] (d), [25°C H₂O 41.0 mg+23.0 mg] (e), and [25°C CH₃OH 41.0 mg+23.0 mg] (f). According to these results, it is found that the solvents could observably influence the crystal structures (see e and f), which could induce the selective ion transport through these CPOS materials.

We further performed the zeta potential measurements of these CPOS materials as shown in Figure R8. It is found that temperature has a slight effect on zeta potential. At high temperature, the crystal becomes uneven, showing different surface charges (a-c). When using high concentration of the materials (d), the zeta potential decreases, which is possibly due to the excessive -NH₂. Notably, the effect of solvents on zeta potential is obvious (e, f), which endows the CPOS materials with the smaller zeta potential. This is possibly attributed to the effect of solvents on the exposed -SO₃⁻ in CPOS pores. Besides, the proportion of exposed crystal face might lead to different surface charge. In sum, the growth condition used in the manuscript is ideal and optimal, conducive to the crystal growth in the PI nanochannel with higher -SO₃⁻ exposures for high-performance cation transport.

Figure R8. Zeta potential of CPOS obtained in different conditions. As shown in (a-c), zeta potential of CPOS obtained in different temperatures is similar. If increasing the concentration of materials (d), zeta potential decreases possibly due to the more -NH_2 groups crosslinked in pores of CPOS. If using other solvents such as H_2O and CH_3OH (e, f), zeta potential observably decreases due to the effect of solvents on the exposed -SO_3^- in CPOS pores.

According to the CO_2 adsorption-desorption isotherms, all of these CPOS materials show a type-I isotherm followed by a sharp uptake at low pressures, which is characteristic of microporous materials (Figure R9a-f). The pore size distribution of these CPOS materials was calculated using non-local density functional theory, indicating pores sizes of 10.5 \AA (Figure R9g). Further insight can be found that the distribution of pore diameter for CPOS material in this work shows a single shoulder peak at 7.9 \AA ; however, other CPOS materials obtained using different synthesis conditions show disordered peaks from 4 to 9 \AA , indicating other CPOS materials could have unstable pore structure which is not good for precise and selective ion transport. These results also indicate that the CPOS material obtained in this work is the optimal.

Figure R9. The CO₂ adsorption-desorption isotherms of the CPOS materials and distribution of their pore diameter. [25°C THF 41.0 mg+23.0 mg] (a), [40°C THF 41.0 mg+23.0 mg] (b), [55°C THF 41.0 mg+23.0 mg] (c), [25°C THF 82.0 mg+46.0 mg] (d), [25°C H₂O 41.0 mg+23.0 mg] (e), and [25°C CH₃OH 41.0 mg+23.0 mg] (f); distribution of pore diameter of these CPOS materials (g).

More importantly, we measured the ion transport and sieving behaviors of CPOS materials obtained in different conditions. As shown in Figure R10, we could find that all of the CPOS materials show selective K⁺ ion transport, which have high K⁺ ion flux up to ~100 ppm in permeation solutions. In detail, CPOS used in this work (Figure R10a) displays the highest monovalent ion sieving such as K⁺/Li⁺, K⁺/Na⁺, and K⁺/Rb⁺ selectivity of ~20, ~61, and ~113, respectively. However, other CPOS materials display the lower ion selectivity (Figure R10b-f), which could be attributed to the weaker surface charge and unsatisfactory crystal morphology in the PI nanochannel. These deficiencies lead to inadequate ion selectivity; for example, the concentrations of Li⁺, Na⁺, and Rb⁺ ion increase through other CPOS materials as shown in Figure R10b-f and result in the strongly competitive penetration in mixed solutions. Although the ion selectivity of these materials (Figure R10b-f) is lower than that of the materials used in this work (Figure R10a), they still maintain the obvious monovalent ion separation properties, indicating the universality of ion sieving by this kind of materials. Based on

these results, we selected the optimal synthesis condition for the separation of monovalent ions from these conditions.

Figure R10. Ion transport behavior and ion sieving performance of different CPOS materials obtained in [25°C THF 41.0 mg+23.0 mg] (a), [40°C THF 41.0 mg+23.0 mg] (b), [55°C THF 41.0 mg+23.0 mg] (c), [25°C THF 82.0 mg+46.0 mg] (d), [25°C H $_2$ O 41.0 mg+23.0 mg] (e), and [25°C CH $_3$ OH 41.0 mg+23.0 mg] (f).

Comment 2: It is difficult to understand the necessity of the conical morphology of the nanopore for the high selectivity of the current system, since the origin of the selectivity was explained based on the structure of the nanochannel formed in the CPOS microcrystal and the hydration/dehydration of the transported ions. Without proving the importance of the conical shape of the nanochannel, it is not suitable to claim the similarity of the current design with KcsA channel.

Response: We appreciate the reviewer for the professional comments. Potassium ion channels in living organisms have asymmetric structures and can open and close to control ion transport (*Nature* 2001, 414, 43-48). In addition, biological ion channels have the ability to unidirectionally pass ions. Siwy *et al.* proposed the essence of ion rectification in nanochannel (*Adv. Funct. Mater.* 2006, 16, 735-746) and a synthetic electrical field-driven ion pump based on a conical nanochannel to pump ions (*Phys. Rev. Lett.* 2002, 89, 198103), which could help researchers to achieve the biomimetic functions (*Chem. Soc. Rev.* 2010, 39, 1115-1132). For example, by designing the charged asymmetric nanochannel, we successfully achieve unidirectional ion transport performance under the external bias (for example, negative bias for conduction, positive bias for closure, as shown in Figure R11). Besides, the recently published papers show that the construction of asymmetric nanochannels is a useful way to achieve biomimetic channels (*Chem. Soc. Rev.* 2017, 47, 322-356; *J. Am. Chem. Soc.* 2015, 137, 6011-6017; *Sci. Adv.* 2016, 2, e1600689). We also used cylindrical nanochannel filled with CPOS with same synthesis method as comparison, and found that, the function of unidirectional ion transport could not be achieved (Figure R11). More importantly, the conical nanochannel show the

“off” state to avoid the filtrate solution backward transport to feed solution, ensuring high-performance ion sieving and diffusion. In that case, the non-conical nanochannel shows the bidirectional ion transport and diffusion. As the reviewer said, there is no one-to-one matching relationship between the conical shape and KcsA channel, which we strongly agree with.

Figure R11. Conical nanochannel achieves the unidirectional ion transport from negative bias to positive bias. More importantly, the conical nanochannel show the “off” state to avoid the filtrate solution backward transport to feed solution, ensuring high-performance ion sieving and diffusion. In that case, the non-conical nanochannel shows the bidirectional ion transport and diffusion.

Comment 3: The list of precedent systems summarized in Supplementary Table 2 is quite informative for the researchers in relevant community and would be appreciated. Based on this, the authors emphasized that the current system revealed the highest K^+ selectivity over Li^+ and Na^+ compared with other precedent systems. However, the nanopores reported in ref #9 had a higher Na^+/K^+ selectivity than the current system. In addition it is simply true that Li^+/K^+ and Li^+/Na^+ were not reported in #9, and this doesn't mean that the authors' system is superior to that reported in #9.

Response: We appreciate the reviewer very much for the comments. The ref. 9 shows the biomimetic potassium-selective nanopores decorated with 4'-aminobenzo-18-crown-6 ether and single-stranded DNA molecules located at one pore entrance, which achieves a high K^+/Na^+ selectivity up to ~80. This work is pioneering and provides an emerging strategy for the construction of biomimetic ion channels/pores. In the section of Introduction, we summarized and highlighted the important work, which not only help us better understand ion transport features in confinement, but also is quite informative for the researchers. In Supplementary Table 2, this work (ref. 9) was also included, which have the highest K^+/Na^+ selectivity among them. To avoid misunderstandings, we delete some sentences and only highlight that our system achieves high-performance K^+ transport and the selectivity of K^+/Li^+ reaches the top level.

For example, “Therefore, the K^+/Li^+ selectivity was calculated to exceed 363, reaching the top level among most artificial nanochannel membranes reported thus far.”

Comment 4: The authors claimed that the high K^+ selectivity was originated from the low hydration energy of K^+ compared with other cations. If so, is the mobility of Rb^+ is higher than K^+ ?

Response: We appreciate the reviewer very much for the comments. We performed the ion permeation experiments to show the selectivity of Rb^+ in a mixture (100 mM $LiCl$, $NaCl$, KCl , $RbCl$,

and CsCl as the feed solutions) through the CPOS pores. The other cell was filled with MilliQ water as the permeation solution. The ion concentration in the permeation was obtained by the inductively coupled plasma mass spectrometry (ICP-MS) measurements. As shown in Figure R12a, the CPOS pores have a selective permeation with a rank of $K^+ \gg Na^+ > Li^+ > Rb^+$, which show the ultra-selective K^+ transport. Based on the permeation results, the ion selectivity of K^+/Rb^+ is up to ~ 110 , which is an excellent advance in the monovalent ion sieving (Figure R12b). Although Rb^+ ($75.5 \text{ kcal mol}^{-1}$) has a lower hydration energy than that of K^+ ($80.9 \text{ kcal mol}^{-1}$), the difference of the energy is slight (Table R2). In addition, Rb^+ has a larger ionic radius and radius of the first hydration shell than that of K^+ , generating a higher transport resistance which could lead to the difficulty for Rb^+ to enter and pass the CPOS pores. Therefore, as shown in Figure R12c, the mobility rate of K^+ is two orders of magnitude higher than that of Rb^+ through the CPOS pores. In this case, we propose that the size effect could play a more key role in the selective transport. Additionally, the ion selectivity of K^+/Li^+ and K^+/Na^+ drops due to the competitive penetration in quinary mixed solutions (*Sci. Adv.* 2020, 6, eabd9045; *Angew. Chem. Int. Ed.* 10.1002/anie.202108801). These contents have been added in Supplementary Information.

Figure R12. (a) Ion permeation through the CPOS pores under an external bias of -1 V . (b) Ion selectivity of the CPOS pores. (c) Mobility rate of K^+ and Rb^+ through the CPOS pores.

Table R2. Ion-dependent parameters including the ionic radius, hydration ionic radius, number of the first hydration shell, and hydration energy.

Ion	Ionic radius (Å)	Hydration ionic radius (Å)	Radius (R_{\min}) of the first hydration shell (Å)	Hydration energy (kcal mol^{-1}) in kT
Li^+	0.60	3.82	2.08	122.2 ± 0.6
Na^+	0.95	3.58	2.36	98.8 ± 0.8
K^+	1.33	3.31	2.80	80.9 ± 1.0
Rb^+	1.48	3.29	2.89	75.5 ± 0.9

Comment 5: Why did K^+ show the highest activation energy for the ion transport among Li^+ , Na^+ and

K⁺, although it reveals the lowest hydration energy?

Response: We appreciate the reviewer very much for the comments. We found that this previous Figure 3f should be replaced, because a mistake occurred while calculating the conductivity according to the Arrhenius formula (the trend is correct as shown in the data of Figure 3f). In order to make the data accurate, we conducted the test again and took the average result for 11 consecutive tests (Figure R13). We have updated this new figure. In the revised Figure, we can find that K⁺ shows the lowest activation and hydration energy (the trend of new results is also consistent with that in previous manuscript).

Figure R13. Activation energy of Li⁺, Na⁺, and K⁺ ion through CPOS pores (also see the revised Figure 3f in the manuscript).

Comment 6: Why did the divalent cations show the ion flux at the positive voltages while the monovalent ones did not as shown in Supplementary Figure 7a?

Response: We appreciate the reviewer very much for the comments. We think that the ion flux at the positive voltages could be generated from the contribution of anion (Cl⁻). We used the same concentration solutions but the concentrations of Cl⁻ are different. For example, the concentration of Cl⁻ in MgCl₂ and CaCl₂ is twice that in LiCl, NaCl, and KCl. Therefore, the ion flux of divalent ion solutions is higher than that of monovalent ion solutions. In addition, the difference of ion flux might come from the influence of competitive penetration due to different cations. These discussions have been added into the Supplementary Information.

Comment 7: CPK model would be helpful for understanding the size of the nanochannel shown in Fig. 1 (iii).

Response: We appreciate the reviewer very much for the suggestions. We have supplemented the CPK model as Figure 1a, iii, also shown in Figure R14.

Figure R14. CPK model for showing the size of the nanochannel with a dimension of $5.3 \times 6.8 \text{ \AA}^2$ (C, grey; N, blue; O, red; S, yellow; H, white).

Reviewer #3 (Remarks to the Author):

The manuscript submitted by Li et al. presents a strategy to construct porous subnanometer structures displaying selective K^+ transport. This strategy relied on the use of a covalent organic framework hosted in an asymmetric nanochannel.

The set of experiments is interesting, but I consider that this work lacks novelty. The use of metal-organic frameworks and covalent organic frameworks hosted in nanochannels for controlling ion transport has been previously reported by the same research group:

- Fast and selective fluoride ion conduction in sub-1-nanometer metal-organic framework channels, Li et al. Nat. Comm. 2019.

- Unidirectional and Selective Proton Transport in Artificial Heterostructured Nanochannels with Nano-to-Subnano Confined Water Clusters, Li et al. Adv. Mater, 2020.

- Ultrasensitive Monovalent Metal Ion Conduction in a Three-Dimensional Sub-1 nm Nanofluidic Device Constructed by Metal–Organic Frameworks, Lu et al. ACS Nano, 2021.

This work describes the exploration of a covalent organic framework that has not been previously reported in prior studies; however, no real advance is reported. For example, one key parameter of these hybrid membranes is the rectification efficiency. Values reported in this manuscript are close to 20 at best (Fig. 2b). This value is well below rectification efficiency values reported almost 15 years ago (Vlassioux and Siwy, Nano Letters, 2007). Given the lack of novelty, I do not recommend the publication of this paper in Nature Communications.

Response: We appreciate the reviewer for comments. First of all, we appreciate the reviewer's positive comments including "The set of experiments is interesting" and "This work describes the exploration of a covalent organic framework that has not been previously reported in prior studies". The strategy of constructing porous subnanometer structures in nanoconfined spacing has been reported in some works such as these listed literatures, which is widely to achieve the bioinspired ion channels. Note that the major contribution in our work is the elucidation of nanochannels-confined CPOS in achieving biomimetic potassium-selective ionic device. The construction of biomimetic potassium-selective ionic device is the most important contribution of our study to the fields of biomimetic nanodevices and nanofluidics and to our best knowledge, our results of ion sieving are

top-of-the-line so far.

In the mentioned works, Xingya Li, et al. (*Wang's group from Monash University, Australia*) used some MOF-based nanofluidic system for the selective F^- conduction, proton transport, and mono-/di-valent ion sieving, wherein the polymer nanochannel was considered as a promising platform for constructing confined space (*Nat. Mater.* 2020, 19, 701-702.). These efforts contribute to controlled ion transport; however, efficient monovalent ion separation remains a challenge.

Our work (*Wen's group from Technical Institute of Physics and Chemistry, CAS, China*) indicates that biological systems are promising sources for the formation of subnanometer channels for the transport of molecules and ions, wherein the electrostatic distribution of charges on the channel walls and steric factors play a strategic role for building useful functions. By analogy, we have engineered a permanently porous 3D molecular architecture, namely crystalline porous organosulfonate-amidinium salts (CPOS) instead of COF or MOF materials, containing sub-nm channels formed by a double helix with two intertwined and alternatively charged ribbons, characteristic of transmembrane channels. In order to obtain the functions similar to natural KcsA channel with selective K^+ transport, CPOS materials with abundant active sites were filled in PI nanochannel to mimic potassium-channel proteins. Measurement results showed that the functional nanochannel indeed achieves ultra-selective K^+ ion transport with high monovalent ion sieving of the K^+/Li^+ and K^+/Na^+ selectivity ratios of 363 and 31, respectively. These performances are at the top level compared with the currently reported results.

To our best knowledge, different from the obvious discrepancy in the valence and dimension between monovalent and divalent ions, K^+ , Na^+ , and Li^+ (they can be widely found in sea sources) ions have the same valence and similar size. Thus, these parameters lead to the difficulty in separating monovalent ions, which has become a challenge and hot subject for scientists. In addition, materials with the high-performance sieving such as K^+/Li^+ and K^+/Na^+ could be widely used in water desalination and selective ion electrode, solvent dehydration, potassium-based batteries and supercapacitors and molecular sieving. Therefore, our work could represent a step towards monovalent ion sieving and a promising prototype of a synthetic membrane channel valuable for bionanofluidic.

This paper (*Nano Lett.* 2007, 7, 552-556) is a pioneer work that focuses on the control of ion rectification and its mechanism study. In our work, the ion rectification can be achieved due to the conical nanochannel that helps the system to conduct unidirectional ion transport under the external bias. In fact, the ion rectification is not the focus of this work, but rather an auxiliary aspect of mimicking natural ion channels with unidirectional ion conduction. High rectification can be achieved by reducing the size of the tip, but making it difficult for the crystal growth in the PI nanochannel. Besides, our goal is to achieve high-efficient and unidirectional ion sieving in nanochannels. In this work, ion selectivity could be obtained by ICP-MS measurements rather than ion rectification. According to the ICP-MS results, this nanochannel shows ultrahigh K^+/Li^+ and K^+/Na^+ selectivity, providing an effective methodology for creating *in vitro* biomimetic devices with high-performance K^+ ion sieving.

Additionally, we also provide other significances of our work compared with other works:

- (1) We proposed that the ultrahigh ion sieving stems from the synergistic effects of cation- π and electrostatic interactions, resulting in a higher energy barrier for Li^+ and Na^+ and selective K^+ transport, which could provide new avenues for analyzing other selective ion transport processes;
- (2) In some supplementary experiments, we found that the CPOS pores achieved the ultrahigh K^+/Cs^+ selectivity up to ~ 1300 , which could provide potential in radioactive element separation such as

cesium isotopes. This work certainly suggests the potential of artificial ion channels based on CPOS materials towards water desalination and metal ion trace detection. We believe that advances in porous materials with asymmetry and specific functions integrated into hierarchical structures will lead to new opportunities, including synthetic ion pump system and anisotropic thermally conductive materials.

REVIEWER COMMENTS

Reviewer #1 (Remarks to the Author):

Review report for “Biomimetic KcsA channels with ultra-selective K⁺ transport for monovalent ion sieving”

The authors have satisfactorily addressed many of my comments, but some of my questions regarding the conductivity calculation were left unanswered (see Comment 2 in the rebuttal letter). Moreover, I believe the notation and some of the calculations are not correct.

The authors have explained that they apply the conductivity model to the CPOS pores and not to the nanochannel in the PI membrane (this issue was extremely confusing in the first manuscript). Now, I agree that the model can be used for large (> 1 M) ionic strengths, where the double layer is smaller than the diameter of CPOS pore. Its validity under lower ionic strengths is still questionable, but the authors do not use it in this limit, so I see no conflict there. There are, however, still other important issues with this section:

1) In my original review, I asked how the conductivity was calculated. The authors replied that they determine it from the equation $G = I/U \times l/S$, where I is the ionic current at voltage U , l is the length of the conical channel and S is the cross-sectional area. First of all, the use of G for conductivity (units S/cm) is very misleading because G is always used for conductance (units of S). Conductivity is typically represented by the Greek letter kappa (κ).

In the caption of Figure 2c (and in the rebuttal letter to my comment #2), the authors use "conductance" instead of "conductivity". Please correct the caption.

Please check the standard symbols and definitions here:
[https://en.wikipedia.org/wiki/Conductivity_\(electrolytic\)](https://en.wikipedia.org/wiki/Conductivity_(electrolytic))

The answer provided by the authors to my original question is unsatisfactory because the nanochannels rectify current (note that we are discussing the behavior of the channels, not of the CPOS pores). Therefore, the value of kappa will strongly depend on the choice of U . The only correct choice is to use a very small value of U , where the system has an ohmic (linear) behavior (i.e., $I = U \cdot R$). For this reason, G and kappa should be determined from the slope of the I vs U plot near $U = 0$. It is not enough to use always the same voltage (as the authors mention in their response). Otherwise, the authors should prove that their conclusions hold regardless of the choice of U .

2) The authors indicate that they calculate S (the cross-sectional area) in the equation from the diameter of the base of the channel (D_{base}). That calculation is only correct for a cylindrical channel. The resistance of a truncated conical channel is well known (see Nuclear Instruments and Methods in Physics Research Section B: Beam Interactions with Materials and Atoms, 184(3), 337–346, doi:10.1016/s0168-583x(01)00722-4). The problem is basically analogous to that of finding the electrical resistance of a truncated conical conductor.

The proper equation is (doi:10.1016/s0168-583x(01)00722-4):

$$G = I/U = \pi \cdot \kappa \cdot D \cdot d / (4 L)$$

where G is the conductance (units of S), κ is the conductivity (units of S/cm), D is the

cone diameter, d is the tip diameter, and L is the length of the channel. As expected, this equation depends on the diameters of both the base and the tip, not only the base one.

Therefore, the authors should calculate the conductivity using the following equation:

$$\kappa = G / (\pi * D * d) * 4 * L = I / U * (\pi * D * d) * 4 * L$$

This change in the calculation will only introduce a multiplicative factor, but that factor is important for the following question.

3) In my original review, I asked how the fitting parameters determined from the fit in Figure 2c (conductivity vs salt concentration) compare with experiments. That question was left unanswered. In the present application of the formula, the dimensions of the CPOS pores are known, so the authors can obtain the sum of the ion mobilities ($\mu^+ + \mu^-$) inside the pore and compare them with the corresponding bulk values. This calculation is very important to convincingly show that model is applicable.

Reviewer #2 (Remarks to the Author):

The authors have made sincere efforts to address the comments raised by this and other reviewers. In particular, the experimental evidence, that the quality of crystals, highly influenced by crystallization conditions, is important for high ion selectivity, is very convincing. These results give important information for the researchers in relevant fields and are also worth publishing elsewhere. Now the manuscript has been suitably revised with additional experimental data and discussions to support the authors' claims. So, I recommend this manuscript to be published in Nature Communications.

Response to the comments of the reviewers

We appreciate the constructive comments and suggestions from the reviewers, and we have revised the manuscript accordingly as detailed in the responses below. The corresponding changes have been highlighted in yellow in the main text.

Reviewer #1 (Remarks to the Author):

Comment: The authors have satisfactorily addressed many of my comments, but some of my questions regarding the conductivity calculation were left unanswered (see Comment 2 in the rebuttal letter). Moreover, I believe the notation and some of the calculations are not correct.

Response: We appreciate the Reviewer #1's valuable comments and suggestions which significantly improve our manuscript. Meanwhile, we apologize for the erroneous notation and calculations. We have carefully revised these contents according to the Reviewer #1's suggestions. Furthermore, we also provided responses to comments point-by-point and revised the manuscript, as follows.

Comment 1: The authors have explained that they apply the conductivity model to the CPOS pores and not to the nanochannel in the PI membrane (this issue was extremely confusing in the first manuscript). Now, I agree that the model can be used for large ($> 1 \text{ M}$) ionic strengths, where the double layer is smaller than the diameter of CPOS pore. Its validity under lower ionic strengths is still questionable, but the authors do not use it in this limit, so I see no conflict there. There are, however, still other important issues with this section:

In my original review, I asked how the conductivity was calculated. The authors replied that they determine it from the equation $G = I/U \times l/S$, where I is the ionic current at voltage U , l is the length of the conical channel and S is the cross-sectional area. First of all, the use of G for conductivity (units S/cm) is very misleading because G is always used for conductance (units of S). Conductivity is typically represented by the Greek letter kappa (κ).

In the caption of Figure 2c (and in the rebuttal letter to my comment #2), the authors use "conductance" instead of "conductivity". Please correct the caption.

Please check the standard symbols and definitions here: [https://en.wikipedia.org/wiki/Conductivity_\(electrolytic\)](https://en.wikipedia.org/wiki/Conductivity_(electrolytic))

Response: Thanks very much for the reviewer's professional suggestions. We have uniformly used "conductance" in Figure 2c and revised the corresponding descriptions to avoid any misunderstandings. In addition, we redefined "conductivity" which is represented by " κ " in Methods for calculating the ion conductivity at different temperatures (as shown Figure 3f). We are also grateful to the reviewer for providing us with some reference information, which help us better understand some standard symbols and definitions.

Comment 2: The answer provided by the authors to my original question is unsatisfactory because the nanochannels rectify current (note that we are discussing the behavior of the channels, not of the CPOS pores). Therefore, the value of kappa will strongly depend on the choice of U . The only correct choice is to use a very small value of U , where the system has an ohmic (linear) behavior (i.e., $I = U \cdot R$). For this reason, G and kappa should be determined from the slope of the I vs U plot near $U = 0$.

It is not enough to use always the same voltage (as the authors mention in their response). Otherwise, the authors should prove that their conclusions hold regardless of the choice of U.

Response: Thanks very much for the reviewer's professional comments. Just as the reviewer indicated, in the calculation of conductance, we have tried different voltages and found a voltage, which is small enough to show the ohmic behavior. For example, in conductance tests, we adopted a smaller step voltage (0.02 V) to obtain I-V curves. We found that the fitted curve is linear when the voltage is less than 0.2 V. Therefore, the used voltage was set to 0.1 V, where the system has an ohmic behavior, allowing us to calculate the ion conductance by I/U. These details have been also added in the revised manuscript.

Comment 3: The authors indicate that they calculate S (the cross-sectional area) in the equation from the diameter of the base of the channel (D_{base}). That calculation is only correct for a cylindrical channel. The resistance of a truncated conical channel is well known (see Nuclear Instruments and Methods in Physics Research Section B: Beam Interactions with Materials and Atoms, 184(3), 337–346, doi:10.1016/s0168-583x(01)00722-4). The problem is basically analogous to that of finding the electrical resistance of a truncated conical conductor. The proper equation is (doi:10.1016/s0168-583x(01)00722-4): $G = I/U = \pi * \kappa * D * d / (4 L)$, where G is the conductance (units of S), kappa is the conductivity (units of S/cm), D is the cone diameter, d is the tip diameter, and L is the length of the channel. As expected, this equation depends on the diameters of both the base and the tip, not only the base one. Therefore, the authors should calculate the conductivity using the following equation: $\kappa = G / (\pi * D * d) * 4 * L = I / U * (\pi * D * d) * 4 * L$. This change in the calculation will only introduce a multiplicative factor, but that factor is important for the following question.

Response: Thanks for the reviewer's significant suggestions which help us further improve this manuscript and make our results much more accurate. In Figure 2c, the equation $G=10^3(\mu_+ + \mu_-)c_b N_A e w h / l + 2\mu_+ \sigma w / l$ is generally used to describe the ion conductance (I/U) behavior in a single channel or pore. To match the data in Figure 2c with the conductance equation and avoid misunderstandings, we therefore revised Figure 2c and used the conductance-concentration curve. In addition, we quite agree with the suggestion from Reviewer on the new equation. According to the equation $\kappa = 4LG / (\pi D_{base} d_{tip})$, we recalculated the values of ion conductivities at different temperatures also provided the updated Figure 3f. These details and related literature (ref. 51, Nucl. Instrum. Methods Phys. Res., Sect. B 2001, 184, 337-346.) have been added in Methods of the revised manuscript. These revised figures are shown below (the new Figure 2c and Figure 3f as Figure R1 and Figure R2, respectively).

Figure R1 (new Figure 2c). Transmembrane ion conductance as a function of the solution

concentration.

Figure R2 (new Figure 3f). Temperature dependence of the ion conductivity for of Li⁺ (green square), Na⁺ (purple circle), and K⁺ (pink hexagon). Solid symbols represent experimental data; dashed lines represent curve-fitting results.

Comment 4: In my original review, I asked how the fitting parameters determined from the fit in Figure 2c (conductivity vs salt concentration) compare with experiments. That question was left unanswered. In the present application of the formula, the dimensions of the CPOS pores are known, so the authors can obtain the sum of the ion mobilities ($\mu_+ + \mu_-$) inside the pore and compare them with the corresponding bulk values. This calculation is very important to convincingly show that model is applicable.

Response: Thanks very much for the reviewer’s comments. We apologize for not providing satisfying answers to this question in the first round of response. In details, in high concentration (≥ 1 M) solutions, the thickness of electric double layer (EDL) is far thinner than the width of the CPOS pore, where ion transport is regarded as bulk-like ion transport without the effect of surface charge of CPOS pore. To verify such behaviors, linear fitting of ionic conductance in high concentration was carried out, and the fitting degree reached 99.5%. The results show that the ionic conductance inside the pores is linear with high concentration (1M, 2M, and 3M). As known, the conductance of ion transport in bulk is also linear with ion concentration. Therefore, it is also experimentally proved that the ion conductance in CPOS pores is bulk-like behavior in high concentration. In sum, the fitting line indicates that the ionic conductance inside the pores at high concentration is bulk-like behavior.

Additionally, in low concentration solution, the ion conductance is mainly contributed by $2\mu_+\sigma w/l$ where the value of ($\mu_+ + \mu_-$) cannot be calculated. If in high concentration, the ($\mu_+ + \mu_-$) value is assigned to the bulk transport ($G=10^3(\mu_++\mu_-)c_b N_A ewh/l$), which does not represent ion transport behavior inside pores. Therefore, this formula is more suitable as a qualitative result. In order to obtain the value of ($\mu_+ + \mu_-$), we performed the drift-diffusion experiments to obtain the sum of the ion

mobilities: $(\mu_+ + \mu_-)_{\text{inside}} = 1.16 \times 10^{-7} \text{ m}^2 \text{ V}^{-1} \text{ S}^{-1}$. Compared with the bulk value of $(\mu_+ + \mu_-)_{\text{bulk}} = 1.55 \times 10^{-7} \text{ m}^2 \text{ V}^{-1} \text{ S}^{-1}$, the lower $(\mu_+ + \mu_-)$ value inside pores is attributed to the ultralow μ_- through the negatively charged pores. We calculated the μ_+ and μ_- in CPOS pores and bulk, respectively, and the results show that μ_+ in CPOS pores is $9.76 \times 10^{-8} \text{ m}^2 \text{ V}^{-1} \text{ S}^{-1}$, which is higher than $7.62 \times 10^{-8} \text{ m}^2 \text{ V}^{-1} \text{ S}^{-1}$ in bulk; on the contrary, μ_- in CPOS pores is $1.87 \times 10^{-8} \text{ m}^2 \text{ V}^{-1} \text{ S}^{-1}$, which is lower than $7.91 \times 10^{-8} \text{ m}^2 \text{ V}^{-1} \text{ S}^{-1}$ in bulk. This also indicates the CPOS pores can enhanced cation transport and reduce anion transport. These results are also shown in Figure 3i.

Reviewer #2 (Remarks to the Author):

The authors have made sincere efforts to address the comments raised by this and other reviewers. In particular, the experimental evidence, that the quality of crystals, highly influenced by crystallization conditions, is important for high ion selectivity, is very convincing. These results give important information for the researchers in relevant fields and are also worth publishing elsewhere. Now the manuscript has been suitably revised with additional experimental data and discussions to support the authors' claims. So, I recommend this manuscript to be published in Nature Communications.

Response: We appreciate the reviewer's high comments and recommendation.

REVIEWER COMMENTS

Reviewer #1 (Remarks to the Author):

The authors have addressed all my questions and made the required changes to the manuscript. In my opinion, the manuscript can be published in its current form.

Response to the comments of the reviewers

Reviewer #1 (Remarks to the Author):

Comments: The authors have addressed all my questions and made the required changes to the manuscript. In my opinion, the manuscript can be published in its current form.

Response: Thanks very much for the reviewer's comments.